# Closing the gap between SVRG and TD-SVRG with Gradient Splitting

## Abstract

Temporal difference (TD) learning is a simple algorithm for policy evaluation in reinforcement learning. The performance of TD learning is affected by high variance and it can be naturally enhanced with variance reduction techniques, such as the Stochastic Variance Reduced Gradient (SVRG) method. Recently, multiple works have sought to fuse TD learning with SVRG to obtain a policy evaluation method with a geometric rate of convergence. However, the resulting convergence rate is significantly weaker than what is achieved by SVRG in the setting of convex optimization. In this work we utilize a recent interpretation of TD-learning as the splitting of the gradient of an appropriately chosen function, thus simplifying the algorithm and fusing TD with SVRG. We prove a geometric convergence bound with predetermined learning rate of 1/8, that is identical to the convergence bound available for SVRG in the convex setting.

## 1 Introduction

Reinforcement learning (RL) is a learning paradigm which addresses a class of problems in sequential decision making environments. Policy evaluation is one of those problems, which consists of determining expected reward agent will achieve if it chooses actions according to stationary policy. Temporal Difference learning (TD learning, Sutton (1988)) is popular algorithm, since it is simple and might be performed online on single samples or small mini-batches. TD learning method uses Bellman equation to bootstrap the estimation process update the value function from each incoming sample or minibatch. As all methods in RL, TD learning from the "curse of dimensionality" when number of states is large. To address this issue, in practice linear or nonlinear feature approximation of state values is often used.

Despite its simple formulation, theoretical analysis of approximate TD learning is subtle. There are few important milestones in this process, one of which is a work of Tsitsiklis & Van Roy (1997), in which asymptotic convergence guarantees were established. More recently advances were made by Bhandari et al. (2018), Srikant & Ying (2019) and Liu & Olshevsky (2020). In particular, the last paper shows that TD learning might be viewed as an example of gradient splitting, a process analogous to gradient descent.

TD-learning has inherent variance problem, which is that the variance of the update does not go to zero as the method converges. This problem is also present in a class of convex optimization problems where target function is represented as a sum of functions and SGD-type methods are applied Robbins & Monro (1951). Such methods proceed incrementally by sampling a single function, or a minibatch of functions, to use for stochastic gradient evaluations. A few variance reduction techniques were developed to address this problem and make convergence faster, including SAG Schmidt et al. (2013), SVRG Johnson & Zhang (2013) and SAGA Defazio et al. (2014). These methods are collectively known as *variance-reduced gradient methods*. The distinguishing feature of these methods is that they converge geometrically.

The first attempt to adapt variance reduction to TD learning with online sampling was done by Korda & La (2015). Their results were discussed by Dalal et al. (2018) and Narayanan & Szepesvári (2017); Xu et al. (2020) performed reanalysis of their results and shown geometric convergence for Variance Reduction Temporal Difference learning (VRTD) algorithm for both Markovian and i.i.d sampling. The work of Du et al. (2017) directly apply SVRG and SAGA to a version of policy

evaluations by transforming it into an equivalent convex-concave saddle-point problem. Since their algorithm uses two sets of parameters, in this paper we call it Primal Dual SVRG or PD SVRG.

All these results obtained geometric convergence of the algorithms, improving the sub-geometric convergence of the standard TD methods. However, the convergence rates obtained in these papers are significantly worse than the convergence of SVRG in convex setting. In particular, the resulting convergence times for policy evaluations scaled with the square of the condition number, as opposed to SVRG which retains the linear scaling with the condition number of SGD. Quadratic scaling makes practical application of theoretically obtain values almost unfeasible, since number of computations becomes very large even for simple problems. Moreover, the convergence time bounds contained additional terms coming from the condition number of a matrix that diagonalizes some of the matrices appearing the problem formulations, which can be arbitrarily large.

In this paper we analyze the convergence of the SVRG technique applied to TD (TD-SVRG) in two settings: $(i)$ a pre-sampled trajectory of the *Markov Decision Process (MDP)* (finite sampling), and $(ii)$ when states are sampled directly from the MDP (online sampling). Our contribution is threefold:

- For the finite sample case we achieve significantly better results with simpler analysis. We are first to show that TD-SVRG has the same convergence rate as SVRG in the convex optimization setting with a pre-determined learning rate of 1/8.

- For i.i.d. online sampling, we similarly achieve better results with simpler analysis. Similarly, we are first to show that TD-SVRG has the same convergence rate as SVRG in the convex optimization setting with a predetermined learning rate of 1/8. In addition, for Markovian online sampling, we provide convergence guarantees that in most cases are better than state-of-the art results.

- We are the first to develop theoretical guarantees for an algorithm that can be directly applied to practice. In previous works, batch sizes required to guarantee convergence were very large that made them impractical (see Subsection H.1) and grid search was needed to optimize the learning rate and batch size values. We include experiments that show our theoretically obtained batch size and learning rate can be applied in practice and achieve geometric convergence.

## 2 PROBLEM FORMULATION

We consider a discounted reward Markov Decision Process (MDP) $(\mathcal{S}, \mathcal{A}, \mathcal{P}, r, \gamma)$, where $\mathcal{S}$ is a state space, $\mathcal{A}$ is an action space, $\mathcal{P} = \mathcal{P}(s'|s, a)_{s, s' \in \mathcal{S}, a \in \mathcal{A}}$ are the transition probabilities, $r = r(s, s')$ are the rewards and $\gamma \in [0, 1)$ is a discount factor. In this MDP agent follows policy $\pi$, which is a mapping $\pi : \mathcal{S} \times \mathcal{A} \to [0, 1]$. Given that policy is fixed, for the remainder of the paper we will consider transition matrix $P$, such that: $P(s, s') = \sum_a \pi(s, a)\mathcal{P}(s'|s, a)$. We assume, that Markov process produced by transition matrix is irreducible and aperiodic with stationary distribution $\mu_\pi$.

The policy evaluation problem is to compute $V^\pi$, defined as: $V^\pi(s) := E\left[\sum_{t=0}^\infty \gamma^t r_{t+1}\right]$. Here $V^\pi$ is the value function, formally defined to be the unique vector which satisfies the equality $T^\pi V^\pi = V^\pi$, where $T^\pi$ is a Bellman operator, defined as: $T^\pi V^\pi(s) = \sum_{s'} P(s, s')\left(r(s, s') + \gamma V^\pi(s')\right)$. The TD(0) method is defined as follows: one iteration performs a fixed point update on randomly sampled pair of states $s, s'$ with learning rate $\eta$: $V(s) \leftarrow V(s) + \eta(r(s, s') + \gamma V(s') - V(s))$. When the state space size $|\mathcal{S}|$ is large, tabular methods which update a value for every state $V(s)$ become impractical. For this reason linear approximation is often used. Each state a represented as feature vector $\phi(s) \in \mathbb{R}^d$ and state value $V^\pi(s)$ is approximated by $V^\pi(s) \approx \phi(s)^T \theta$, where $\theta$ is a tunable parameter vector. Now a single TD update on randomly sampled transition $s, s'$ becomes:

$$\begin{aligned} \theta &\leftarrow \theta + \eta g_{s,s'}(\theta) \\ &= \theta + \eta((r(s, s') + \gamma\phi(s')^T\theta - \phi(s)^T\theta)\phi(s)), \end{aligned}$$

where the second equation should be viewed as a definition of $g_{s,s'}(\theta)$.

Our goal is to find parameter vector $\theta^*$ such that average update vector is zero

$$\mathbb{E}_{s,s'}[g_{s,s'}(\theta^*)] = 0.$$

This expectation is also called mean-path update $\bar{g}(\theta)$ and can be written as:

Table 1: Algorithms parameter comparison. PD SVRG and PD SAGA results reported from Du et al. (2017), VRTD and TD results from Xu et al. (2020), GTD2 from Touati et al. (2018). $\lambda_{\min}(Q)$ and $\kappa(Q)$ are used to define, respectively, minimum eigenvalue and condition number of matrix $Q$. $\lambda_A$ in this table denotes minimum eigenvalue of matrix $1/2(A + A^T)$. Other notation is taken from original papers. For simplicity $1 + \gamma$ is upper bounded by 2.

| Setting | Method | Learning rate | Batch size | Total complexity |
|---------|--------|---------------|------------|------------------|
| Finite sample | GTD2 | $\frac{9^2 \times 2\sigma}{8\sigma^2(k+2)+9^2\zeta}$ | 1 | $\mathcal{O}(\frac{\kappa(Q)^2\mathcal{H}d}{\lambda_{\min}(G)\epsilon})$ |
| | PD SVRG | $\frac{\lambda_{\min}(A^TC^{-1}A)}{48\kappa(C)L_G^2}$ | $\frac{51\kappa^2(C)L_G^2}{\lambda_{\min}(A^TC^{-1}A)^2}$ | $\mathcal{O}(\frac{\kappa^2(C)L_G^2}{\lambda_{\min}(A^TC^{-1}A)^2}\log(\frac{1}{\epsilon}))$ |
| | PD SAGA | $\frac{\lambda_{\min}(A^TC^{-1}A)}{3(8\kappa C^2 L_G^2+n\mu_\rho)}$ | 1 | $\mathcal{O}(\frac{\kappa^2(C)L_G^2}{\lambda_{\min}(A^TC^{-1}A)^2}\log(\frac{1}{\epsilon}))$ |
| | This paper | $1/8$ | $16/\lambda_A$ | $\mathcal{O}(\frac{1}{\lambda_A}\log(\frac{1}{\epsilon}))$ |
| i.i.d sampling | TD | $\min(\frac{\lambda_A}{16}, \frac{2}{\lambda_A})$ | 1 | $\mathcal{O}(\frac{1}{\epsilon\lambda_A^2}\log(\frac{1}{\epsilon}))$ |
| | VRDT | $\lambda_A/64$ | $\frac{132}{\lambda_A^2}$ | $\mathcal{O}(\max(\frac{1}{\epsilon}, \frac{1}{\lambda_A^2})\log(\frac{1}{\epsilon}))$ |
| | This paper | $1/8$ | $16/\lambda_A$ | $\mathcal{O}(\max(\frac{1}{\epsilon}, \frac{1}{\lambda_A})\log(\frac{1}{\epsilon}))$ |

$$
\begin{aligned}
\bar{g}(\theta) &= \mathbb{E}_{s,s'}[g_{s,s'}(\theta)] \\
&= \mathbb{E}_{s,s'}[(\gamma\phi(s')^T\theta - \phi(s)^T\theta)\phi(s)] + \mathbb{E}_{s,s'}\left[r(s,s')\phi(s)\right] \\
&:= -A\theta + b,
\end{aligned}
$$

where the last line should be taken as the definition of $A$ and $b$. Finally, the minimum eigenvalue of matrix $(A + A^T)/2$ plays an important role in our analysis and will be denoted as $\lambda_{\min}$.

There are few possible setting of the problem: the samples $s, s'$ might be drawn from the MDP on-line (Markovian sampling) or independently (*i.i.d.* sampling): first state $s$ is drawn from $\mu_\pi$, then $s'$ is drawn from correspondent row of $P$. The latter case analysis is covered in 6. Another possible setting for analysis is the "finite sample set" setting, in which states a data set $\mathcal{D} = \{(s_t, a_t, r_t, s_{t+1})\}_{t=1}^N$ of size $N$ is drawn ahead of time following Markov sampling, and TD(0) proceeds by drawing samples from this data set. We analyze this case in Sections 4 and 5.

We make following standard assumptions:

**Assumption 1.** *The matrix $A$ is non-singular.*

**Assumption 2.** *$||\phi(s)||_2 \leqslant 1$ for all $s \in \mathcal{S}$.*

Assumption 1 needed to guarantee that $A^{-1}b$ exists and the problem is solvable. Assumption 2 is introduced for simplicity, it always might be fulfilled by rescaling feature matrix.

## 2.1 KEY IDEA OF THE ANALYSIS

In our analysis we often use function $f(\theta)$, defined as:

$$f(\theta) = (\theta - \theta^*)^T A(\theta - \theta^*). \tag{1}$$

Function $f(\theta)$ is a key characteristic function of TD learning. In their paper Liu & Olshevsky (2020) introduce function $f(\theta)$ as $(1 - \gamma)||V_\theta - V_{\theta*}||^2_D + \gamma||V_\theta - V_{\theta*}||^2_{Dir}$. Then, authors define gradient splitting (linear function $h(x) = B(x - a)$ is **gradient splitting** of quadratic function $j(x) = (x - a)^T Q(x - a)$, where $Q$ is symmetric positive semi-definite matrix, if $B + B^T = 2Q$) and show that negation to mean-path update $-\bar{g}(\theta)$ is indeed a gradient splitting of function $f(\theta)$. In this paper we do not use the fact that function $f(\theta)$ might be represented as weighted sum of $D$-norm and Dirichlet norm and, for convenience, define function $f(\theta)$ based on its gradient splitting properties.

We rely on interpretation of TD learning as splitting of the gradient descent in our analysis. In Xu et al. (2020) authors note: "In Johnson & Zhang (2013) , the convergence proof relies on the relationship between the gradient and the value of the objective function, but there is not such an objective function in the TD learning problem." Well, viewing on TD learning as gradient splitting gives the relationship between the gradient and the value function, which allows implementation of similar analysis as in Johnson & Zhang (2013) to achieve stronger results. It also gives the objective function $f(\theta)$ which is better measure of the distance to optimal solution, rather than $||\theta - \theta*||^2$, and yields tighter convergence bounds.

## 3    THE TD-SVRG ALGORITHM

We next propose a modification of the TD(0) method (TD SVRG) which can attain a geometric rate. This algorithm is given below as Algorithm 1. The algorithm works under the "fixed sample set" setting which assumes there already exists a sampled data set $\mathcal{D}$. This is the same setting was considered in Du et al. (2017). However, the method we propose does not add regularization and does not use dual parameters, which makes it considerably simpler.

---

**Algorithm 1** TD-SVRG for finite sample case

> **Parameters** update frequency $M$ and learning rate $\eta$
> **Initialize** $\tilde{\theta}_0$.
> **Iterate:** for $m = 1, 2, \ldots$
> > $\theta = \tilde{\theta}_{m-1}$,
> > $\bar{g}(\theta) = \frac{1}{N} \sum_{s,s' \in \mathcal{D}} g_{s,s'}(\theta)$,
> > where $g_{s,s'}(\theta) = (r(s, s') + \gamma\phi(s')^T\theta - \phi(s)^T\theta)\phi(s_t)$,
> > $\theta_0 = \theta$.
> > **Iterate:** for $t = 1, 2, \ldots, M$
> > > Randomly sample $s, s'$ from the dataset and compute update vector
> > > $v_t = g_{s,s'}(\theta_{t-1}) - g_{s,s'}(\theta) + \bar{g}(\theta)$.
> > > Update parameters $\theta_t = \theta_{t-1} - \eta v_t$.
> > **end**
> > Set $\tilde{\theta}_m = \theta_t$ for randomly chosen $t \in (0, \ldots, M - 1)$.
> **end**

---

Like the classic SVRG algorithm, our proposed TD-SVRG has two layers of loops. We refer one step of the outer loop as an *epoch* and one step of inner loop as an *iteration*. TD-SVRG keeps two parameter vectors: current parameter vector $\theta_t$, which is being updated every iteration, and the vector $\tilde{\theta}$, which is updated the end of each epoch. In the beginning of outer loop, the mean-path TD update vector $\bar{g}(\tilde{\theta})$ is computed with a pass through the entire data set. This vector is used in inner loop to compute local updates $v_t = g_{s,s'}(\theta_{t-1}) - g_{s,s'}(\tilde{\theta}) + \bar{g}(\theta)$, where $g_{s,s'}(\theta_{t-1})$ is a TD update computed on a uniformly randomly sampled data point from $\mathcal{D}$ with current parameter vector $\theta_t$ and $g_{s,s'}(\tilde{\theta})$ is a TD update computed on the same data point. Each iteration ends with an update of epoch vector, which is randomly chosen from parameter vectors during the epoch.

## 4    CONVERGENCE ANALYSIS

In this section we show, that under simple assumption Algorithm 1 attain geometric convergence in terms of specially chosen function $f(\theta)$ with $\eta$ is $\mathcal{O}(1)$ and $M$ is $\mathcal{O}(1/\lambda_{\min})$.

## 4.1 PRELIMINARIES

In order to analyze the convergence of the presented algorithm we define expected square norm of difference in current and optimal parameters as $w(\theta)$ :

$$w(\theta) \quad = \quad E_{s,s'}||g_{s,s'}(\theta) - g_{s,s'}(\theta^*)||^2. \tag{2}$$

With this notation we provide an technical lemma. All of our proofs are based on variations of this lemma.

**Lemma 1.** *If Assumptions 1, 2 hold, epoch parameters of two consecutive epochs $m-1$ and $m$ are related by the following inequality:*

$$2\eta M \mathbb{E} f(\tilde{\theta}_m) - 2M\eta^2 \mathbb{E} w(\tilde{\theta}_m) \leqslant \mathbb{E}||\tilde{\theta}_{m-1} - \theta^*||^2 + 2\eta^2 M \mathbb{E} w(\tilde{\theta}_{m-1}), \tag{3}$$

*where the expectation is taken with respect to all previous epochs and choices of states $s, s'$ during the epoch $m$.*

*Proof.* The proof of the lemma generally follows the analysis logic in Johnson & Zhang (2013), it might be found in Appendix A. □

Lemma 1 plays an auxiliary role in our analysis and significantly simplifies it. It introduces a new approach to the convergence proof by carrying iteration to iteration and epoch to epoch bounds to the earlier part of the analysis. In particular, deriving bounds in terms of some arbitrary function $u(\theta)$ is now reduced to deriving upper bounds on $||\tilde{\theta}_{m-1}||^2$ and $w(\theta)$ and a lower bound on $f(\theta)$ in terms of the function $u$. In fact, the function $f(\theta)$ itself will play the role of $u(\theta)$ in our proof.

In addition, it is now easy to demonstrate the point we made in Subsection 2.1. The main problem of direct application of the SVRG convergence analysis to TD learning is that it requires the target function $P(w)$ to be a sum of convex functions $\phi_i(w)$, where all functions $\phi_i$ are $L$-Lipschitz and $\gamma$-smooth (notation here is from Johnson & Zhang (2013)). Of course, the TD learning problem cannot be represented in this form. However, the main use of $L$-Lipschitz and smoothness properties in the original is to derive a bound on the expected norm of the difference between current and optimal parameters. As will be shown later, in TD learning this expected norm ($w(\theta)$ in our notation) still might be derived, even when a sum representation does not exist.

## 4.2 WARM-UP: CONVERGENCE IN TERMS OF SQUARED NORM

Firstly, we derive the convergence bound in terms of $||\theta - \theta^*||^2$ and show that they are consistent with previous results.

**Proposition 1.** *Suppose Assumptions 1, 2 hold. If we chose learning rate as $\eta = \lambda_{\min}/32$ and number of iteration as $M = 32/\lambda_{\min}^2$, then Algorithm 1 has a convergence rate of:*

$$E[||\tilde{\theta}_m - \theta^*||^2] \leqslant \left(\frac{5}{7}\right)^m ||\tilde{\theta}_0 - \theta^*||^2.$$

*Proof.* The proof is given in Appendix B □

Note that deriving a convergence rate in terms of squared norm $||\tilde{\theta}_m - \theta^*||^2$ leads to batch size $m$ to be $\mathcal{O}(1/\lambda_{min}^2)$, which is better than results in Du et al. (2017), since their results has complexity $\mathcal{O}(\kappa^2(C)\kappa_G^2)$, where $\kappa(C)$ is condition number of matrix $C = \mathbb{E}_{s \in \mathcal{D}}[\phi(s)\phi(s)^T]$ and $\kappa_G \propto 1/\lambda_{\min}(A^T C^{-1} A)$. Experimental comparison of these values is provided in Subsection H.1 .

### 4.3 First Main Result: Convergence in terms of $f(\theta)$

In this section derive a bound in terms of $f(\theta)$. For analysis to be simpler and illustrative we introduce one more assumption which we ease in section 4.5.

**Assumption 3** (Dataset Balance). *In the dataset $\mathcal{D}$ first state of the first sample and the second state of the last sample is the same state, i.e. $s_1 = s_{N+1}$.*

We need this assumption to omit dataset bias, so that states $s$ and $s'$ have the same distribution, as in the original MDP. Having this assumption, we can proof theorem 1:

**Theorem 1.** *Suppose Assumptions 1, 2, 3 hold. If we choose learning rate $\eta = 1/8$ and number of inner loop iterations $M = 16/\lambda_{\min}$, then Algorithm 1 will have a convergence rate of:*

$$E[f(\tilde{\theta}_m)] \leqslant \left(\frac{2}{3}\right)^m f(\tilde{\theta}_0).$$

Note that $\tilde{\theta}_m$ refers to the iterate after $m$ iterations of the outer loop. Because we choose the length $M$ of the inner loop to be $16/\lambda_{\min}$, the total number of samples guaranteed by this theorem until $E[f(\tilde{\theta}_m)] \leqslant \epsilon$ is actually $(16/\lambda_{\min})\log(1/\epsilon)$.

*Proof of Theorem 1.* The proof is given in Appendix C. ∎

Convergence analysis without Assumption 3 provided in Appendix Section D.

### 4.4 Similarity of SVRG and TD-SVRG

Liu & Olshevsky (2020) show that negation to mean-path update $-\bar{g}(\theta)$ is a gradient splitting of $f(\theta)$. In this work we show even greater importance of function $f(\theta)$ for TD learning process. Recall convergence rate obtained in Johnson & Zhang (2013) for sum of convex functions setting:

$$\frac{1}{\gamma\eta(1 - 2L\eta)m} + \frac{2L\eta}{1 - 2L\eta},$$

where $\gamma$ is a strong convexity parameter and $L$ is Lipschitz smoothness parameter (employing notation from the original paper). Function $f(\theta) = (\theta - \theta^*)^T A(\theta - \theta^*)$ is $2\lambda_{min}(A)$ strongly convex and 2-Lipschitz smooth, which means that convergence rate obtained in this paper is identical to the convergence rate of SVRG in convex setting (we have slightly better bound $L$ instead of $2L$ due to strong bound on $w(\theta)$ we derived for this setting). This fact further extends the analogy between TD learning and convex optimization earlier explored by Bhandari et al. (2018) and Liu & Olshevsky (2020).

## 5 Batching SVRG Case Analysis

In this section we extend our results to inexact mean-path update computation, applying the results of Babanezhad et al. (2015) to TD SVRG algorithm. We show that geometric convergence rate might be achieved with smaller number of computations by estimating mean-path TD-update instead of performing full computation. This approach is similar to Peng et al. (2019), but again doesn't require introduction of dual variables. In addition, we provide a particular way to compute $n_m$, which might be used in practice.

Since computation of mean-path error is not related to the dataset balance, in this section for simplicity we assume that dataset is balanced.

**Theorem 2.** *Suppose Assumptions 1, 2, 3 hold, then if learning rate is chosen as $\eta = 1/8$ number of inner loop iterations $M = 16/\lambda_{\min}$ and batch size $n_m = \min(N, \frac{N}{c\rho^{2m}(N-1)}(2|r_{max}|^2 + 8||\tilde{\theta}_m||^2))$, where $c$ is a parameter, Algorithm 2 will have a convergence rate of:*

$$E[f(\tilde{\theta}_m)] \leqslant \rho^m(f(\tilde{\theta}_0) + C),$$

*where $\rho \in (0, 1)$ is convergence rate and $C$ is some constant.*

---

**Algorithm 2** TD-SVRG with batching for finite sample case

    **Parameters** update frequency $M$ and learning rate $\eta$
    **Initialize** $\tilde{\theta}_0$.
    **Iterate:** for $m = 1, 2, \ldots$
        $\theta = \tilde{\theta}_{m-1}$,
        choose batch size $n_m$,
        sample batch $\mathcal{D}^m$ of size $n_m$ from $\mathcal{D}$ without replacement,
        compute $\mu = \frac{1}{n_m} \sum_{s,s' \in \mathcal{D}^m} g_{s,s'}(\theta)$,
        where $g_{s,s'}(\theta) = (r(s, s') + \gamma \phi(s')^T \theta - \phi(s)^T \theta)\phi(s_t)$,
        $\theta_0 = \tilde{\theta}$.
        **Iterate:** for $t = 1, 2, \ldots, M$
            Randomly sample $s, s'$ from $\mathcal{D}$ and compute update vector
            $v_t = g_{s,s'}(\theta_{t-1}) - g_{s,s'}(\tilde{\theta}) + \mu$,
            Update parameters $\theta_t = \theta_{t-1} - \eta v_t$.
        **end**
        set $\tilde{\theta}_m = \theta_t$ for randomly chosen $t \in (0, \ldots, M-1)$.
    **end**

---

*Proof.* The proof is given in Appendix E. □

This theorem shows that during early epochs approximation of mean-path update is good enough to guarantee geometric convergence. However, the batch size used for approximation increases geometrically with each epoch with rate $\rho^2$, where $\rho$ is a desired convergence rate, until it reaches size of the dataset $N$. The constant $C$ depends on parameter $c$ and upper bound $Z = \max_\theta(|\theta - \theta^*|)$, where the max is taken over all parameter vectors seen during the run of the algorithm.

# 6    Second main result: Online iid sampling from the MDP

In this section we apply TD learning as gradient splitting analysis to the case of online i.i.d sampling from the MDP each time we need to generate a new state $s$. We show that this view of TD learning as gradient splitting might be applied in this case to derive tighter convergence bounds. One issue of TD-SVRG in i.i.d. setting is that mean-path update may not be computed directly. However, this issue might be addressed with sampling technique introduced in Section 5, which makes i.i.d. case very similar to TD-SVRG with non-exact mean-path computation in finite samples case.

In this setting, geometric convergence is clearly not attainable with variance reduction, which always relies on a pass through the entire dataset. Since here there is no data set, and one samples from the MDP at every step, one clearly cannot make a pass through all states of the MDP (or, rather, this is unrealistic to do in practice). To obtain convergence, one needs to take increasing batch sizes. Our next theorem does so, while improving the scaling with condition number from quadratic to linear.

TD-SVRG algorithm for iid sampling case is very similar to Algorithm 2, with only difference that states $s, s'$ are being sampled from the MDP instead of the dataset $\mathcal{D}$. Formal definition of Algorithm 3 might be found in section F.

**Theorem 3.** *Suppose Assumptions 1, 2 hold, then if learning rate is chosen as $\eta = 1/8$, number of inner loop iterations $M = 16/\lambda_{\min}$ and batch size $n_m = \frac{1}{c\rho^{2m}}(2|r_{max}|^2 + 8||\theta||^2)$, where $c$ is some arbitrary chosen constant, Algorithm 3 will have a convergence rate of:*

$$E[f(\tilde{\theta}_m)] \leqslant \rho^m(f(\tilde{\theta}_0) + C),$$

*where $\rho \in (0, 1)$ is convergence rate and $C$ is some constant.*

*Proof.* The proof is given in Appendix F. □

To parse this, observe that as in 5, each epoch requires $n_m$ computations to estimate mean-path update and $16/\lambda_{\min}$ to perform inner loop iterations. Thus, as epoch number $m$ grows, $n_m$ will

dominate $16/\lambda_{\min}$, which results in the total computational complexity of $\mathcal{O}(\max(\frac{1}{\epsilon}, \frac{1}{\lambda_{\min}}) \log(\frac{1}{\epsilon}))$, which is better than $\mathcal{O}(\max(\frac{1}{\epsilon}, \frac{1}{\lambda_{\min}^2}) \log(\frac{1}{\epsilon}))$ shown by Xu et al. (2020) when $\lambda_{\min} > \epsilon$.

In practice, $\lambda_{\min}$ is determined by the MDP and the feature matrix, while $\epsilon$ is a desired accuracy, thus, the former is given and the latter might be chosen. In most scenarios, $\lambda_{\min}$ is a small number and $\epsilon$ is chosen such that $\epsilon < \lambda_{\min}$. Even if this is not a case, $\lambda_{\min}^2$ is a very small number and most likely $\epsilon < \lambda_{\min}^2$. Thus, in the absolute majority of cases results shown in this paper are stronger.

The same convergence result with predetermined constant learning rate cannot be derived for Markovian sampling case, in which update sampling strategy is different from one in classical SVRG sampling. However, gradients splitting interpretation of TD learning still allows to achieve better convergence guarantees than in previous works for the absolute majority of problems. Algorithm, discussion and convergence proof is provided in Appendix Section G.

## 7 EXPERIMENTS

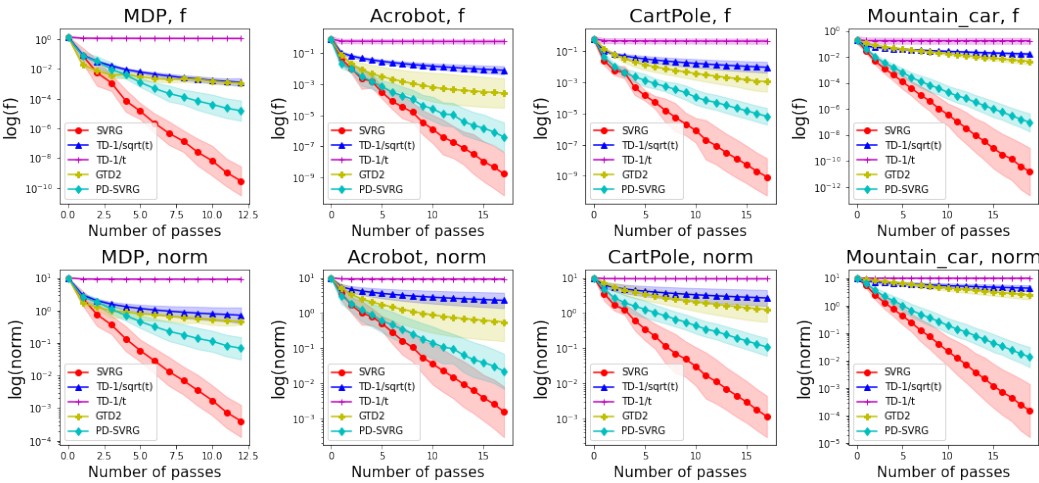

Figure 1: Average performance of different algorithms in finite sample case. Columns - dataset source environments: MDP, Acrobot, CartPole and Mountain Car. Rows - performance measurements: $\log(f(\theta))$ and $\log(|\theta - \theta^*|)$.

### 7.1 ALGORITHMS COMPARISON

In this set of experiments we compare the performance of TD-SVRG with GTD2 Sutton et al. (2009), Vanilla TD learning Sutton (1988) and PD SVRG Du et al. (2017) in finite sample setting. Generally, experiment set-up is similar to Peng et al. (2019). Datasets of size 5000 are generated from 4 environments: Random MDP Dann et al. (2014) and Acrobot, CartPole and Mountain car OpenAI Gym environments Brockman et al. (2016). For Random MDP, we construct MDP environment with $|S| = 400$, 21, features and 10 actions, with actions choice probabilities generated from $U[0, 1)$. For OpenAI gym environments, agent select states uniformly at random. Features constructed by applying RBF kernels to original states and then removing highly correlated features (correlation coefficient $> 0.5$). To produce datasets of similar sizes we resampled dataset if smallest eigen-value of its matrix $A$ was outside the interval $[0.32, 0.54] \cdot 10^{-4}$, which corresponds to TD-SVRG batch sizes between 30000 and 50000. Decay rate $\gamma$ is set to 0.95.

Hyperparameters for algorithms selected as follows: for TD-SVRG theoretically justified parameters are selected, learning rate $\eta = 1/8$ and number of inner loop computations $M = 16/\lambda_{\min}$; for GTD2 we used parameters which are suggested for small problems $\alpha = 1$ and $\beta = 1$. For vanilla TD decreasing learning rates are set to $\alpha = 1/\sqrt{t}$ and $\alpha = 1/t$. For PD-SVRG setting parameters to theoretically suggested is not feasible, since even for simple problems values of number of inner

loop computations $M$ is too large (see Appendix Subsection H.1). Following original paper Du et al. (2017) we run a simple grid search are pick best performing values, which are $\sigma_\theta = \sigma_w = 0.1/\lambda_{\max}(\hat{C})$. Results presented on Figure 1. Each algorithm for each setting was run 10 times, average result is presented. As theory predicts, TD-SVRG and PD-SVRG converge geometrically, while GTD and vanilla TD converge sub linearly.

## 7.2 ONLINE IID SAMPLING FROM THE MDP

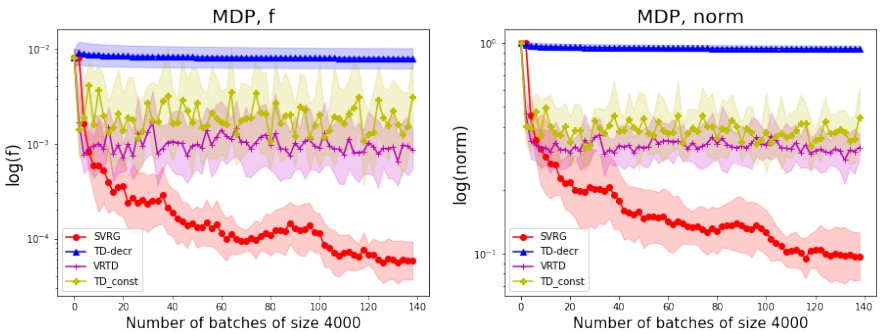

Figure 2: Average performance of TD-SVRG, VRTD and vanilla TD in i.i.d. sampling case. "TD-decr" refers to vanilla TD with decrasing learning rate, "TD-const" - to vanilla TD with constant learning rate. Left figure - performance in terms of $\log(f(\theta))$, right in terms of $\log(|\theta - \theta*|)$.

In this set of experiments we compare the performance of TD-SVRG, VRTD and three Vanilla TD with fixed and decreasing learning rates in i.i.d. sampling case. States and rewards are sampled from the same MDP as in Section 7.1. Hyperparameters are chosen as follows: for TD-SVRG - learning rate $\eta = 1/8$, number of inner loop computations $M = 16/\lambda_{\min}$. VRTD - learning rate $\alpha = 0.1$ and batch size $M = 2000$. For vanilla TD with constant learning rate its value set to $0.1$ and for decreasing learning rate it is $1/t$, where $t$ is number of performed update. Average results over 10 runs presented on Figure 2.

## 7.3 REPRODUCIBILITY

Authors provide a link to anonymous github repository with code and instruction how to reproduce the experiments.

## 8 CONCLUSION

In the paper we utilize a view on TD learning as splitting of gradient descent to show that SVRG technique applied to TD updates attain similar convergence rate as SVRG in convex function setting. Our analysis addresses both finite sample and i.i.d. sampling cases, which previously were analyzed separately, and improves state of the art bounds in both cases. In addition we show that gradient splitting interpretation helps to improve convergence guarantees in Markovian sampling case. The algorithms based on our analysis have fixed learning rate and small number of inner loop computation, easy to implement and demonstrates good performance during experiments.

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

# A    PROOF OF LEMMA 1

The proof follows the same logic as in Johnson & Zhang (2013) and organized in four steps.

**Step A.1.** *In the original paper proof starts with deriving a bound on the squared norm of the difference between current and optimal sets of parameters. Since the introduction of $w(\theta)$ this step in our proof is trivial.*

$$E_{s,s'}||g_{s,s'}(\theta) - g_{s,s'}(\theta^*)||^2 = w(\theta)$$

**Step A.2.** *During Step 2 we derive a bound on the norm of an single iteration $t$ update $v_t = g_{s,s'}(\theta_{t-1}) - g_{s,s'}(\tilde{\theta}) + \bar{g}(\tilde{\theta})$, assuming that states $s, s'$ were sampled randomly during step $t$:*

$$
\begin{aligned}
E_{s,s'}[||v_t||^2] &= \mathbb{E}||g_{s,s'}(\theta_{t-1}) - g_{s,s'}(\tilde{\theta}) + \bar{g}(\tilde{\theta})||^2 \\
&= E_{s,s'}||(g_{s,s'}(\theta_{t-1}) - g_{s,s'}(\theta^*)) + (g_{s,s'}(\theta^*) - g_{s,s'}(\tilde{\theta}) + \bar{g}(\tilde{\theta})||^2 \\
&\leqslant 2E_{s,s'}||(g_{s,s'}(\theta_{t-1}) - g_{s,s'}(\theta^*))||^2 \\
&\quad + 2E_{s,s'}||g_{s,s'}(\tilde{\theta}) - g_{s,s'}(\theta^*) - (\bar{g}(\tilde{\theta}) - \bar{g}(\theta^*))||^2 \\
&= 2E_{s,s'}||(g_{s,s'}(\theta_{t-1}) - g_{s,s'}(\theta^*))||^2 + 2E_{s,s'}||g_{s,s'}(\tilde{\theta}) - g_{s,s'}(\theta^*) \\
&\quad - E_{s,s'}[g_{s,s'}(\tilde{\theta}) - g_{s,s'}(\theta^*)]||^2 \\
&\leqslant 2E_{s,s'}||(g_{s,s'}(\theta_{t-1}) - g_{s,s'}(\theta^*))||^2 + 2E_{s,s'}||g_{s,s'}(\tilde{\theta}) - g_{s,s'}(\theta^*)||^2 \\
&= 2w(\theta_{t-1}) + 2w(\tilde{\theta})
\end{aligned}
$$

*The first inequality uses $\mathbb{E}||a + b||^2 \leqslant 2\mathbb{E}||a||^2 + 2\mathbb{E}||b||^2$. The second inequality uses the face that second central moment is smaller than second moment. The last equality uses the equality from step 1.*

**Step A.3.** *During this step we derive a bound on the expected squared norm of a distance to optimal parameter vector after a single update $t$:*

$$
\begin{aligned}
\mathbb{E}_{s,s'}||\theta_t - \theta^*||^2 &= \mathbb{E}_{s,s'}||\theta_{t-1} - \theta^* + \eta v_t||^2 \\
&= ||\theta_{t-1} - \theta^*||^2 + 2\eta(\theta_{t-1} - \theta^*)\mathbb{E}v_t + \eta^2\mathbb{E}||v_t||^2 \\
&\leqslant ||\theta_{t-1} - \theta^*||^2 + 2\eta(\theta_{t-1} - \theta^*)\bar{g}(\theta_{t-1}) + 2\eta^2 w(\theta_{t-1}) + 2\eta^2 w(\tilde{\theta}) \\
&= ||\theta_{t-1} - \theta^*||^2 - 2\eta f(\theta_{t-1}) + 2\eta^2 w(\theta_{t-1}) + 2\eta^2 w(\tilde{\theta})
\end{aligned}
$$

*The inequality uses the bound obtained in step 2. After rearranging terms it becomes:*

$$\mathbb{E}||\theta_t - \theta^*||^2 + 2\eta f(\theta_{t-1}) - 4\eta^2 w(\theta_{t-1}) \leqslant ||\theta_{t-1} - \theta^*||^2 + 4\eta^2 w(\tilde{\theta})$$

**Step A.4.** *During this step we sum the inequality obtained in Step 3 over the epoch and take expectations with respect to all choices of pair of states $s, s'$ and all previous history and use the random choice property to obtain Equation 1 which relates parameter vectors of two consecutive epochs:*

$$\sum_{t=1}^{M} \mathbb{E}||\theta_t - \theta^*||^2 + \sum_{t=1}^{M} 2\eta\mathbb{E}f(\theta_{t-1}) - \sum_{t=1}^{M} 2\eta^2\mathbb{E}w(\theta_{t-1}) \leqslant \sum_{t=1}^{M} \mathbb{E}||\theta_{t-1} - \theta^*||^2 + \sum_{t=1}^{M} 2\eta^2\mathbb{E}w(\tilde{\theta})$$

*We analyze this expression term-wise.*

$\sum_{t=1}^{M} \mathbb{E}||\theta_{t-1} - \theta^*||^2$ *and* $\sum_{t=1}^{M} \mathbb{E}||\theta_t - \theta^*||^2$ *consist of same terms, except the first term in the first sum and the last term in the last sum, which are $\mathbb{E}||\theta_0 - \theta^*||^2$ and $\mathbb{E}||\theta_M - \theta^*||^2$. Since $\mathbb{E}||\theta_M - \theta^*||^2$ is always positive and it is on the left hand side of the inequality, we could drop it.*

*We denote the parameter vector $\theta$ chosen for epoch parameters in the end of the epoch $\tilde{\theta}_m$. Since this vector is chosen uniformly at random among all iteration vectors $\theta_t$, $\sum_{t=1}^{M} \mathbb{E}f(\theta_{t-1}) = M\mathbb{E}f(\tilde{\theta}_m)$ and $\sum_{t=1}^{M} \mathbb{E}w(\theta_{t-1}) = M\mathbb{E}w(\tilde{\theta}_m)$.*

*At the same time, $\tilde{\theta}$, which was chosen in the end of the previous epoch remains the same throughout the epoch, therefore, $\sum_{t=1}^{M} \mathbb{E}w(\tilde{\theta}) = M\mathbb{E}w(\tilde{\theta})$. Note, that current epoch starts with setting $\theta_0 = \tilde{\theta}$. Also, to underline it is previous epoch, we denote it as $\tilde{\theta}_{m-1}$.*

*Plugging this values in we have 3:*

$$2\eta M \mathbb{E}f(\tilde{\theta}_m) - 2M\eta^2 \mathbb{E}w(\tilde{\theta}_m) \leqslant \mathbb{E}||\tilde{\theta}_{m-1} - \theta^*||^2 + 2\eta^2 M \mathbb{E}w(\tilde{\theta}_{m-1})$$

## B  PROOF OF PROPOSITION 1

To transform inequality 3 from Lemma 1 into convergence rate guarantee, we need to bound $w(\theta)$ and $f(\theta)$ in terms of $||\theta - \theta^*||^2$. Both bounds are easy to show:

$$
\begin{aligned}
w(\theta) &= E_{s,s'}||g_{s,s'}(\theta) - g_{s,s'}(\theta^*)||^2 \\
&= (\theta - \theta^*)^T E_{s,s'}[(\gamma\phi(s') - \phi(s))\phi(s)^T \phi(s)(\gamma\phi(s') - \phi(s))^T](\theta - \theta^*) \\
&\leqslant (\theta - \theta^*)^T E_{s,s'}[||(\gamma\phi(s') - \phi(s))|| \cdot ||\phi(s)|| \cdot ||\phi(s)|| \cdot ||(\gamma\phi(s') - \phi(s))||](\theta - \theta^*) \\
&\leqslant 4||\theta - \theta^*||^2, \\
f(\theta) &= (\theta - \theta^*)^T E_{s,s'}[\phi(s)(\phi(s) - \gamma\phi(s'))^T](\theta - \theta^*) \geqslant \lambda_{min}||\theta - \theta^*||^2.
\end{aligned}
$$

Plugging these bounds into Equation 3 we have:

$$(2\eta M\lambda_{min} - 8M\eta^2)||\tilde{\theta}_m - \theta^*||^2 \leqslant (1 + 8M\eta^2)||\tilde{\theta}_{m-1} - \theta^*||^2.$$

Which yields epoch to epoch convergence rate of:

$$\frac{1 + 8M\eta^2}{2\eta M\lambda_{min} - 8M\eta^2}.$$

For this expression to be $< 1$, we need that $\eta M$ to be $\mathcal{O}(1/\lambda_{min})$, which means that $\eta$ needs to be $\mathcal{O}(\lambda_{min})$ for $M\eta^2$ to be $\mathcal{O}(1)$. Therefore, $M$ need to be $\mathcal{O}(1/\lambda_{min}^2)$. Setting $\eta = \lambda_{min}/32$ and $m = 32/\lambda_{min}^2$ yields convergence rate of $5/7$.

## C  PROOF OF THEOREM 1

The same as in the previous section, we start with deriving bounds, but this time we bound $||\theta - \theta^*||^2$ and $w(\theta)$ in terms of $f(\theta)$. First bound is straightforward:

$$f(\theta) = (\theta - \theta^*)^T E_{\phi,\phi'}[\phi(\phi - \gamma\phi')^T](\theta - \theta^*) \implies ||\theta - \theta^*||^2 \leqslant \frac{1}{\lambda_{min}}f(\theta).$$

For $w(\theta)$ we have:

$$w(\theta) = (\theta - \theta^*)^T E_{s,s'}[(\gamma\phi(s') - \phi(s))\phi(s)^T\phi(s)(\gamma\phi(s') - \phi(s))^T](\theta - \theta^*)$$

$$= (\theta - \theta^*)^T \Big[\frac{1}{N}\sum_{s,s'\in\mathcal{D}} (\gamma\phi(s') - \phi(s))\phi^T(s)\phi(s)(\gamma\phi(s') - \phi(s))^T\Big](\theta - \theta^*)$$

$$\leqslant (\theta - \theta^*)^T \Big[\frac{1}{N}\sum_{s,s'\in\mathcal{D}} (\gamma\phi(s') - \phi(s))(\gamma\phi(s') - \phi(s))^T\Big](\theta - \theta^*)$$

$$= (\theta - \theta^*)^T \Big[\frac{1}{N}\sum_{s,s'\in\mathcal{D}} \gamma^2\phi(s')\phi(s')^T - \gamma\phi(s')\phi(s)^T\Big](\theta - \theta^*) + f(\theta) \qquad (4)$$

$$= (\theta - \theta^*)^T \Big[\frac{1}{N}\sum_{s,s'\in\mathcal{D}} \gamma^2\phi(s)\phi(s)^T - \gamma\phi(s)\phi(s')^T\Big](\theta - \theta^*) + f(\theta)$$

$$\leqslant 2f(\theta),$$

first inequality uses Assumption 2, third equality uses Assumption 3, $(\sum_{s'}\gamma^2\phi(s')\phi(s')^T = \sum_s \gamma^2\phi(s)\phi(s)^T$, since $s$ and $s'$ are the same set of states). The last inequality uses the fact that $\gamma < 1$.

Plugging these bound into Equation 3, we have:

$$2\eta M\mathbb{E}f(\tilde\theta_m) - 4M\eta^2\mathbb{E}f(\tilde\theta_m) \leqslant \frac{1}{\lambda_{min}}\mathbb{E}f(\tilde\theta_{m-1}) + 4\eta^2 M\mathbb{E}f(\tilde\theta_{m-1}),$$

which yields epoch to epoch convergence rate of:

$$\mathbb{E}f(\tilde\theta_m) \leqslant \Big[\frac{1}{2\lambda_{min}\eta M(1 - 2\eta)} + \frac{2\eta}{1 - 2\eta}\Big]\mathbb{E}f(\tilde\theta_{m-1}).$$

Setting $\eta = \frac{1}{8}$ and $M = \frac{16}{\lambda_{min}}$ we have the desired inequality.

## D    CONVERGENCE ANALYSIS WITHOUT DATASET BALANCE

In this section we show the convergence bound for the problem without Assumption 3. In this case, the problem is that after the sampling, $s$ and $s'$ in the dataset does not have the same distribution, i.e., the first element of the tuples $(s_t, a_t, r_t, s_{t+1})$ in our dataset $\mathcal{D}$ need not have the same distribution as the last element of this tuple. Indeed, it could happen that a particular state occurs a different numbers of times as the first element of the tuples in $\mathcal{D}$ as compared to the last element, which would not happen under Assumption 3. When this happens, we will say that the data set is *unbalanced*. In that case, $\bar{g}(\theta)$ need not be a gradient splitting of function $f(\theta)$.

One might hope that, when the size of the data set $N$ is large, this effect has an impact which decays to zero with $N$. Our second main result shows something even stronger: we show that the effect of unbalacedness *disappears completely* for large $N$. Thus our next theorem completely recovers the performance attained by Theorem 1 for large $N$. The catch is the size of the dataset has to be at least as big as $\lambda_{\min}^{-1}$ for this to happen.

**Theorem 4.** *Suppose Assumptions 1, 2 hold and dataset is unbalanced. Define error term $J = \frac{4\gamma^2}{N\lambda_{\min}}$. Then, if we choose learning rate $\eta = 1/(8 + J)$ and number of inner loop iterations $M = 2/(\lambda_{\min}\eta)$, Algorithm 1 will have a convergence rate of:*

$$E[f(\tilde\theta_m)] \leqslant \left(\frac{2}{3}\right)^m f(\tilde\theta_0).$$

*Proof.* The proof is given in Appendix D.1. $\qquad\square$

Note, that in this case $\eta \in \mathcal{O}(\frac{1}{\max(1, 1/(N\lambda_{\min}))})$ and $M \in \mathcal{O}(\frac{1}{\lambda_{\min}\eta})$, which always better than parameters required to guarantee convergence of $||\theta - \theta^*||^2$ (Proposition 1). *Note that the guarantees of this theorem are identical to the guarantees of Theorem 1 in the unbalanced case when $N \geqslant \lambda_{\min}^{-1}$.*

## D.1 PROOF OF THEOREM 4

To proof the theorem we follow the same strategy as in C. For the $f(\theta)$ we can use the same bound:

$$f(\theta) = (\theta - \theta^*)^T E_{\phi,\phi'}[\phi(\phi - \gamma\phi')^T](\theta - \theta^*) \implies ||\theta - \theta^*||^2 \leqslant \frac{1}{\lambda_{min}} f(\theta)$$

Bound for $w(\theta)$ is a little bit more difficult:

$$w(\theta) = (\theta - \theta^*)^T \Big[\frac{1}{N} \sum_{s,s'\in\mathcal{D}} (\gamma\phi(s') - \phi(s))\phi^T(s)\phi(s)(\gamma\phi(s') - \phi(s))^T\Big](\theta - \theta^*)$$

$$\leqslant (\theta - \theta^*)^T \Big[\frac{1}{N} \sum_{s,s'\in\mathcal{D}} (\gamma\phi(s') - \phi(s))(\gamma\phi(s') - \phi(s))^T\Big](\theta - \theta^*)$$

$$= (\theta - \theta^*)^T \Big[\frac{1}{N} \sum_{s,s'\in\mathcal{D}} \gamma\phi(s')(\gamma\phi(s') - \phi(s))^T - \phi(s)(\gamma\phi(s') - \phi(s))^T\Big](\theta - \theta^*)$$

$$= (\theta - \theta^*)^T \Big[\frac{1}{N} \sum_{s,s'\in\mathcal{D}} \gamma^2\phi(s')\phi(s')^T - \gamma\phi(s')\phi(s)^T\Big](\theta - \theta^*) + f(\theta)$$

$$= (\theta - \theta^*)^T \Big[\frac{1}{N} \sum_{s,s'\in\mathcal{D}} \gamma^2\phi(s)\phi(s)^T - \gamma\phi(s)\phi(s')^T\Big](\theta - \theta^*) + f(\theta)$$

$$+ \frac{\gamma^2}{N}(\theta - \theta^*)^T(\phi(s_{N+1})\phi(s_{N+1})^T - \phi(s_1)\phi(s_1)^T)(\theta - \theta^*)^T$$

$$\leqslant 2f(\theta) + \frac{\gamma^2}{N}(\theta - \theta^*)^T(\phi(s_{N+1})\phi(s_{N+1})^T - \phi(s_1)\phi(s_1)^T)(\theta - \theta^*)^T.$$

The first inequality follows from the assumption about norms of feature vectors. The third equality is obtained by adding and subtracting $\frac{\gamma^2}{N}(\theta - \theta^*)^T\phi(s_1)\phi(s_1)^T(\theta - \theta^*)$. Second inequality uses the fact that $\gamma^2 < 1$. We denote maximum eigen-value of matrix $\phi(s_{N+1})\phi(s_{N+1})^T - \phi(s_1)\phi(s_1)^T$ as $\mathcal{K}$ (note that $\mathcal{K} \leqslant 1$). Thus,

$$w(\theta) \leqslant 2f(\theta) + \frac{\gamma^2\mathcal{K}}{N}||\theta - \theta^*||^2 \leqslant f(\theta)(2 + \frac{\gamma^2\mathcal{K}}{N\lambda_{min}}) \leqslant \frac{\gamma^2}{N\lambda_{min}}$$

Plugging these bounds into Equation 3 we have:

$$(2\eta M - 2M\eta^2(2 + \frac{\gamma^2}{N\lambda_{min}}))\mathbb{E}f(\tilde{\theta}_m) \leqslant (\frac{1}{\lambda_{min}} + 2\eta^2 M(2 + \frac{\gamma^2}{N\lambda_{min}}))f(\tilde{\theta})$$

Which yields convergence rate of:

$$\frac{1}{\lambda_{min}2\eta M(1 - \eta(2 + \frac{\gamma^2}{N\lambda_{min}}))} + \frac{\eta(2 + \frac{\gamma^2}{N\lambda_{min}})}{1 - \eta(2 + \frac{\gamma^2}{N\lambda_{min}})}$$

To achieve constant convergence rate, for example $\frac{2}{3}$, we set up $\eta$ such that $\eta(2 + \frac{\gamma^2}{N\lambda_{min}}) = 0.25$, thus the second term is equal to 1/3 and $\eta = \frac{1}{8 + \frac{4\gamma^2}{N\lambda_{min}}}$. Then, to make the first term equal to 1/3, we need to set

$$M = \frac{2}{\lambda_{min}\eta} = \frac{2}{\lambda_{min}\frac{1}{8 + \frac{4\gamma^2}{N\lambda_{min}}}}$$

Thus, $\eta$ is a scale of $\frac{1}{\max(1,1/(N\lambda_{min})}$ and $M$ is a scale of $\frac{1}{\lambda_{min}\min(1,N\lambda_{min})}$.

# E  PROOF OF THEOREM 3

In the first part of the proof we derive an inequality which relates model parameters of two consecutive epochs similar to what we achieved in previous proofs, but with error term. In this part of the proof we follow the same 4 steps logic as while proof of Lemma 1. In the second part of the proof we show that there are conditions under which error term converges to 0.

**Step E.1.** *During the first step we use the bound obtained in inequality 4:*

$$w(\theta) \leqslant 2f(\theta)$$

**Step E.2.** *During this step we derive a bound on the squared norm of a single update $\mathbb{E}[||v_t||^2]$. But now, compared to previous case, we do not compute the exact mean-path updated $\bar{g}(\theta)$, but its estimate, and assume our computation has error $\mu = \bar{g}(\theta) + e$. Thus the single update vector will be*

$$v_t = g(\theta_{t-1}) - g(\tilde{\theta}) + \bar{g}(\tilde{\theta}) + e$$

*Thus, the bound on the single update might be derived as:*

$$
\begin{aligned}
\mathbb{E}[||v_t||^2] &= \mathbb{E}||g(\theta_{t-1}) - g(\tilde{\theta}) + \bar{g}(\tilde{\theta}) + e||^2 \\
&= \mathbb{E}||(g(\theta_{t-1}) - \bar{g}(\theta^*)) + (\bar{g}(\theta^*) - g(\tilde{\theta}) + \bar{g}(\tilde{\theta}) + e)||^2 \\
&\leqslant 2\mathbb{E}||(g(\theta_{t-1}) - g(\theta^*))||^2 + 2\mathbb{E}||g(\tilde{\theta}) - g(\theta^*) - (\bar{g}(\tilde{\theta}) - \bar{g}(\theta^*)) - e||^2 \\
&= 2\mathbb{E}||(g(\theta_{t-1}) - g(\theta^*))||^2 + 2\mathbb{E}||g(\tilde{\theta}) - g(\theta^*) - E[g(\tilde{\theta}) - g(\theta^*)] - e||^2 \\
&= 2\mathbb{E}||(g(\theta_{t-1}) - g(\theta^*))||^2 + 2\mathbb{E}||g(\tilde{\theta}) - g(\theta^*) - E[g(\tilde{\theta}) - g(\theta^*)]||^2 \\
&\quad - 2\mathbb{E}\langle g(\tilde{\theta}) - g(\theta^*) - 4E[g(\tilde{\theta}) - g(\theta^*)], e\rangle + 2\mathbb{E}||e||^2 \\
&\leqslant 2\mathbb{E}||(g(\theta_{t-1}) - g(\theta^*))||^2 + 2\mathbb{E}||g(\tilde{\theta}) - g(\theta^*)||^2 + 2\mathbb{E}||e||^2 \\
&= 2w(\theta_{t-1}) + 2w(\tilde{\theta}) + 2\mathbb{E}||e||^2 \\
&\leqslant 4f(\theta_{t-1}) + 4f(\tilde{\theta}) + 2\mathbb{E}||e||^2,
\end{aligned}
$$

*where first inequality uses $\mathbb{E}||A + B||^2 \leqslant 2\mathbb{E}||A||^2 + 2\mathbb{E}||B||^2$, second inequality uses $\mathbb{E}||A - \mathbb{E}[A]||^2 \leqslant \mathbb{E}||A||^2$ and the third inequality uses the result of Step E.1.*

**Step E.3.** *During this step, we derive a bound on a vector norm after a single update:*

$$
\begin{aligned}
\mathbb{E}||\theta_t - \theta^*||^2 &= \mathbb{E}||\theta_{t-1} - \theta^* + (-\eta v_t)||^2 \\
&= ||\theta_{t-1} - \theta^*||^2 - 2\eta(\theta_{t-1} - \theta^*)^T \mathbb{E}v_t + \eta^2 \mathbb{E}||v_t||^2 \\
&\leqslant ||\theta_{t-1} - \theta^*||^2 - 2\eta(\theta_{t-1} - \theta^*)^T \bar{g}(\theta_{t-1}) + 4\eta^2 f(\theta_{t-1}) + 4\eta^2 f(\tilde{\theta}) + 2\eta^2 \mathbb{E}||e||^2 \\
&= ||\theta_{t-1} - \theta^*||^2 - 2\eta(\theta_{t-1} - \theta^*)^T \nabla f(\theta_{t-1}) - 2\eta(\theta_{t-1} - \theta^*)^T e \\
&\quad + 4\eta^2 f(\theta_{t-1}) + 4\eta^2 f(\tilde{\theta}) + 2\eta^2 \mathbb{E}||e||^2
\end{aligned}
$$

*Rearranging terms we obtain:*

$$
\begin{aligned}
&\mathbb{E}||\theta_t - \theta^*||^2 + 2\eta f(\theta_{t-1}) - 4\eta^2 f(\theta_{t-1}) \\
&\leqslant ||\theta_{t-1} - \theta^*||^2 + 4\eta^2 f(\tilde{\theta}) - 2\eta(\theta_{t-1} - \theta^*)^T e + 2\eta^2 \mathbb{E}||e||^2 \\
&\leqslant ||\theta_{t-1} - \theta^*||^2 + 4\eta^2 f(\tilde{\theta}) + 2\eta||\theta_{t-1} - \theta^*|| \cdot ||e|| + 2\eta^2 \mathbb{E}||e||^2
\end{aligned}
$$

**Step E.4.** *Now derive a bound on epoch update. We assume that quantity $||\theta_{t-1} - \theta^*||$ might be bounded by constant $Z$. Similarly, we denote an error term from previous epoch as $e^{m-1}$. We use the similar logic as during the proof of theorem 1. Since error term doesn't change over the epoch, thus, summing over the epoch we have:*

$$\mathbb{E}||\theta_m - \theta^*||^2 + 2\eta M \mathbb{E} f(\tilde{\theta}_m) - 8\eta^2 M \mathbb{E} f(\tilde{\theta}_m) \leqslant$$
$$\mathbb{E}||\theta_0 - \theta^*||^2 + 8\eta^2 M \mathbb{E} f(\tilde{\theta}) + 2M\eta Z \mathbb{E}||e^{m-1}|| + 2\eta^2 M \mathbb{E}||e^{m-1}||^2$$

*Rearranging terms we have the bound:*

$$\mathbb{E} f(\tilde{\theta}_m) \leqslant (\frac{1}{\lambda_{min} 2\eta M(1 - 4\eta)} + \frac{4\eta}{1 - 4\eta}) \mathbb{E} f(\tilde{\theta}_{m-1}) + \frac{1}{1 - 4\eta}(Z \mathbb{E}||e^{m-1}|| + \eta \mathbb{E}||e^{m-1}||^2)$$

To obtain convergence, we need to guarantee geometric convergence of first and second term in the sum separately. The first term is dependent on inner loop updates, its convergence is analyzed in Theorem 1. Here we show how to achieve a similar geometric convergence rate of the second term. Since error term has 0 mean and it is finite sample case with replacement, expected squared norm might be bounded by:

$$\mathbb{E}||e^m||^2 \leqslant \frac{N - n_m}{N n_m} S^2 \leqslant (1 - \frac{n_m}{N})\frac{S^2}{n_m} \leqslant \frac{S^2}{n_m}$$

where $S^2$ is a bound on update vector norm variance. If we want the error to be bounded by $c\rho^{2m}$, we need the number of batch computations $n_m$ to satisfy the condition:

$$n_m \geqslant \frac{S^2}{c\rho^{2m}}$$

Satisfying this condition guarantees that the second term has geometric convergence:

$$\frac{1}{1 - 4\eta}(Z \mathbb{E}||e^{m-1}|| + \eta \mathbb{E}||e^{m-1}||^2) \leqslant \frac{2}{1 - 4\eta} \max(Z\sqrt{c}, \eta c\rho)\rho^m$$

It is only left to derive a bound $S^2$ for on update vector norm sample variance:

$$\frac{1}{N - 1}\sum_{s,s'}||g_{s,s'}(\theta)||^2 - ||\bar{g}(\theta)||^2 \leqslant$$
$$\frac{N}{N - 1}\frac{1}{N}\sum_{s,s'}||g_{s,s'}(\theta)||^2 = \frac{N}{N - 1}\frac{1}{N}\sum_{s,s'}||(r(s, s') + \gamma\phi(s')^T\theta - \phi(s)^T\theta)\phi(s)||^2 \leqslant$$
$$\frac{N}{N - 1}\frac{1}{N}\sum_{s,s'}2||r\phi(s)||^2 + 4||\gamma\phi(s')^T\theta\phi(s)||^2 + 4||\phi(s)^T\theta\phi(s)||^2 \leqslant$$
$$\frac{N}{N - 1}(2|r_{max}|^2 + 4\gamma^2||\theta||^2 + 4||\theta||^2) = \frac{N}{N - 1}(2|r_{max}|^2 + 8||\theta||^2) = S^2$$

## F    PROOF OF THEOREM 4

TD-SVRG algorithm for iid sampling case is described as Algorithm 3:

The proof of its convergence is very similar to E, the only difference is that now we derive expectation with respect to MDP instead of fixed dataset.

---

**Algorithm 3** TD-SVRG for iid sampling case

---

**Parameters** update frequency $M$ and learning rate $\eta$
**Initialize** $\tilde{\theta}_0$.
**Iterate:** for $m = 1, 2, \ldots$
 $\theta = \tilde{\theta}_{m-1}$,
 choose batch size $n_m$,
 sample batch $\mathcal{D}^m$ of size $n_m$,
 compute $\mu = \frac{1}{n_m} \sum_{s,s' \in \mathcal{D}^m} g_{s,s'}(\theta)$,
 where $g_{s,s'}(\theta) = (r(s,s') + \gamma \phi(s')^T \theta - \phi(s)^T \theta)\phi(s_t)$,
 $\theta_0 = \tilde{\theta}$.
 **Iterate:** for $t = 1, 2, \ldots, M$
  Randomly sample $s, s'$ and compute update vector
  $v_t = g_{s,s'}(\theta_{t-1}) - g_{s,s'}(\tilde{\theta}) + \mu$,
  Update parameters $\theta_t = \theta_{t-1} - \eta v_t$.
 **end**
 set $\tilde{\theta}_m = \theta_t$ for randomly chosen $t \in (0, \ldots, M-1)$.
**end**

---

**Step F.1.** *During the first step we use the bound obtained during the proof of theorem 1:*

$$
\begin{aligned}
w(\theta) &= (\theta - \theta^*)^T E_{s,s'}[(\gamma\phi(s') - \phi(s))\phi(s)^T \phi(s)(\gamma\phi(s') - \phi(s))^T](\theta - \theta^*) \\
&= (\theta - \theta^*)^T \Big[ \sum_{s,s'} \mu_\pi(s)P(s,s')(\gamma\phi(s') - \phi(s))\phi^T(s)\phi(s)(\gamma\phi(s') - \phi(s))^T \Big](\theta - \theta^*) \\
&\leqslant (\theta - \theta^*)^T \Big[ \sum_{s,s'} \mu_\pi(s)P(s,s')(\gamma\phi(s') - \phi(s))(\gamma\phi(s') - \phi(s))^T \Big](\theta - \theta^*) \\
&= (\theta - \theta^*)^T \Big[ \sum_{s,s'} \mu_\pi(s)P(s,s')(\gamma^2\phi(s')\phi(s')^T - \gamma\phi(s')\phi(s)^T) \Big](\theta - \theta^*) + f(\theta) \\
&= (\theta - \theta^*)^T \sum_{s,s'} \mu_\pi(s)P(s,s')(\gamma^2\phi(s)\phi(s)^T - \gamma\phi(s)\phi(s')^T) \Big](\theta - \theta^*) + f(\theta) \\
&\leqslant 2f(\theta),
\end{aligned}
\tag{5}
$$

*first inequality uses Assumption 2, third equality uses the fact that $\mu_\pi$ is a stationary distribution of $P$ ($\sum_{s'} \gamma^2 \mu_\pi(s)P(s,s')\phi(s')\phi(s')^T = \sum_{s'} \gamma^2\mu_\pi(s')\phi(s')\phi(s')^T = \sum_s \mu_\pi(s)\gamma^2\phi(s)\phi(s)^T$). The last inequality uses the fact that $\gamma < 1$.*

**Step F.2.** *During this step we derive a bound on the squared norm of a single update $\mathbb{E}[||v_t||^2]$. Similarly with E we assume inexact computation of mean-path update $\mu = \bar{g}(\theta) + e$. Thus the single update vector becomes:*

$$
v_t = g(\theta_{t-1}) - g(\tilde{\theta}) + \bar{g}(\tilde{\theta}) + e
$$

*Norm of this vector is bounded by:*

$$\mathbb{E}[||v_t||^2] = \mathbb{E}||g(\theta_{t-1}) - g(\tilde{\theta}) + \bar{g}(\tilde{\theta}) + e||^2$$
$$= \mathbb{E}||(g(\theta_{t-1}) - \bar{g}(\theta^*)) + (\bar{g}(\theta^*) - g(\tilde{\theta}) + \bar{g}(\tilde{\theta}) + e)||^2$$
$$\leqslant 2\mathbb{E}||(g(\theta_{t-1}) - g(\theta^*))||^2 + 2\mathbb{E}||g(\tilde{\theta}) - g(\theta^*) - (\bar{g}(\tilde{\theta}) - \bar{g}(\theta^*)) - e||^2$$
$$= 2\mathbb{E}||(g(\theta_{t-1}) - g(\theta^*))||^2 + 2\mathbb{E}||g(\tilde{\theta}) - g(\theta^*) - E[g(\tilde{\theta}) - g(\theta^*)] - e||^2$$
$$= 2\mathbb{E}||(g(\theta_{t-1}) - g(\theta^*))||^2 + 2\mathbb{E}||g(\tilde{\theta}) - g(\theta^*) - E[g(\tilde{\theta}) - g(\theta^*)]||^2$$
$$- 2\mathbb{E}\langle g(\tilde{\theta}) - g(\theta^*) - 4E[g(\tilde{\theta}) - g(\theta^*)], e\rangle + 2\mathbb{E}||e||^2$$
$$\leqslant 2\mathbb{E}||(g(\theta_{t-1}) - g(\theta^*))||^2 + 2\mathbb{E}||g(\tilde{\theta}) - g(\theta^*)||^2 + 2\mathbb{E}||e||^2$$
$$= 2w(\theta_{t-1}) + 2w(\tilde{\theta}) + 2\mathbb{E}||e||^2$$
$$\leqslant 4f(\theta_{t-1}) + 4f(\tilde{\theta}) + 2\mathbb{E}||e||^2$$

*where first inequality uses $\mathbb{E}||A + B||^2 \leqslant 2\mathbb{E}||A||^2 + 2\mathbb{E}||B||^2$, second inequality uses $\mathbb{E}||A - \mathbb{E}[A]||^2 \leqslant \mathbb{E}||A||^2$ and the third inequality uses the result of Step F.1.*

**Step F.3.** *Bound on a vector norm after a single update:*

$$\mathbb{E}||\theta_t - \theta^*||^2 = \mathbb{E}||\theta_{t-1} - \theta^* + (-\eta v_t)||^2$$
$$= ||\theta_{t-1} - \theta^*||^2 - 2\eta(\theta_{t-1} - \theta^*)^T \mathbb{E}v_t + \eta^2\mathbb{E}||v_t||^2$$
$$\leqslant ||\theta_{t-1} - \theta^*||^2 - 2\eta(\theta_{t-1} - \theta^*)^T \bar{g}(\theta_{t-1}) + 4\eta^2 f(\theta_{t-1}) + 4\eta^2 f(\tilde{\theta}) + 2\eta^2\mathbb{E}||e||^2$$
$$= ||\theta_{t-1} - \theta^*||^2 - 2\eta(\theta_{t-1} - \theta^*)^T \nabla f(\theta_{t-1}) - 2\eta(\theta_{t-1} - \theta^*)^T e$$
$$+ 4\eta^2 f(\theta_{t-1}) + 4\eta^2 f(\tilde{\theta}) + 2\eta^2\mathbb{E}||e||^2$$

*Rearranging terms we obtain:*

$$\mathbb{E}||\theta_t - \theta^*||^2 + 2\eta f(\theta_{t-1}) - 4\eta^2 f(\theta_{t-1})$$
$$\leqslant ||\theta_{t-1} - \theta^*||^2 + 4\eta^2 f(\tilde{\theta}) - 2\eta(\theta_{t-1} - \theta^*)^T e + 2\eta^2\mathbb{E}||e||^2$$
$$\leqslant ||\theta_{t-1} - \theta^*||^2 + 4\eta^2 f(\tilde{\theta}) + 2\eta||\theta_{t-1} - \theta^*|| \cdot ||e|| + 2\eta^2\mathbb{E}||e||^2$$

**Step F.4.** *Now derive a bound on epoch update. We assume that quantity $||\theta_{t-1} - \theta^*||$ might be bounded by constant $Z$. Similarly, we denote an error term from previous epoch as $e^{m-1}$. We use the similar logic as during the proof of Theorem 1. Since error term doesn't change over the epoch, thus, summing over the epoch we have:*

$$\mathbb{E}||\theta_m - \theta^*||^2 + 2\eta M\mathbb{E}f(\tilde{\theta}_m) - 8\eta^2 M\mathbb{E}f(\tilde{\theta}_m) \leqslant$$
$$\mathbb{E}||\theta_0 - \theta^*||^2 + 8\eta^2 M\mathbb{E}f(\tilde{\theta}) + 2M\eta Z\mathbb{E}||e^{m-1}|| + 2\eta^2 M\mathbb{E}||e^{m-1}||^2$$

*Rearranging terms we have the bound:*

$$\mathbb{E}f(\tilde{\theta}_m) \leqslant \left(\frac{1}{\lambda_{min}2\eta M(1 - 4\eta)} + \frac{4\eta}{1 - 4\eta}\right)\mathbb{E}f(\tilde{\theta}_{m-1}) + \frac{1}{1 - 4\eta}(Z\mathbb{E}||e^{m-1}|| + \eta\mathbb{E}||e^{m-1}||^2)$$

Similarly to C convergence for the first term might be obtained by setting learning rate $\eta = 1/8$ and number of inner loop iterations $M = 16/\lambda_{min}$. To guarantee convergence of the second term, we need to bound $\mathbb{E}||e^m||^2$. In the infinite population with replacement case norm of the error vector is bounded by:

$$\mathbb{E}||e^m||^2 \leqslant \frac{S^2}{n_m}$$

where $S^2$ is a bound update vector norm variance. If we want the error to be bounded by $c\rho^{2m}$, we need the number of batch computations $n_m$ to satisfy the condition:

$$n_m \geqslant \frac{S^2}{c\rho^{2m}}$$

Satisfying this condition guarantees that the second term has geometric convergence:

$$\frac{1}{1-4\eta}(Z\mathbb{E}||e^{m-1}|| + \eta\mathbb{E}||e^{m-1}||^2) \leqslant \frac{2}{1-4\eta}\max(Z\sqrt{c}, \eta c\rho)\rho^m$$

Similarly to E, bound on sample variance $S^2$ might be derived as follows:

$$\sum_{s,s'}\mu_\pi(s)P(s,s')||g_{s,s'}(\theta)||^2 - ||\bar{g}(\theta)||^2 \leqslant$$

$$\sum_{s,s'}\mu_\pi(s)P(s,s')||g_{s,s'}(\theta)||^2 = \sum_{s,s'}\mu_\pi(s)P(s,s')(||(r(s,s') + \gamma\phi(s')^T\theta - \phi(s)^T\theta)\phi(s)||^2) \leqslant$$

$$\sum_{s,s'}\mu_\pi(s)P(s,s')(2||r\phi(s)||^2 + 4||\gamma\phi(s')^T\theta\phi(s)||^2 + 4||\phi(s)^T\theta\phi(s)||^2) \leqslant$$

$$(2|r_{max}|^2 + 4\gamma^2||\theta||^2 + 4||\theta||^2) = (2|r_{max}|^2 + 8||\theta||^2) = S^2$$

## G    MARKOVIAN SAMPLING CASE ALGORITHM AND ANALYSIS.

Markovian sampling case is the hardest to analyse due to its dependence on MDP properties, which makes establishing bounds on various quantities used during the proof much harder. Applying gradient splitting view helps to improve over existing bounds but derived algorithm does not have a nice property of constant learning rate. To deal with sample-to-sample dependencies with utilize one more assumption often used in the literature:

**Assumption 4.** *The considered MDP is irreducible and aperiodic and there exist constant $m > 0$ and $\rho \in (0,1)$ such that*

$$\sup_{s\in S} d_{TV}(\mathbb{P}(s_t \in \cdot|s_0 = s), \pi) \leqslant m\rho^t, \forall t \geqslant 0,$$

*where $d_{TV}(P, Q)$ denotes the total-variation distance between the probability measures P and Q.*

Another thing we need to employ is projection, which will help to set a bound on update vector $v$. Following Bhandari et al. (2018) and Xu et al. (2020) after each iteration we project parameter vector on a ball or radius $R$ (denoted as $\Pi_R(\theta) = \arg\min_{\theta':|\theta'|\leqslant R}|\theta - \theta'|^2$. We assume that $|\theta*| \leqslant R$, choice of $R$ which guarantees it might be found in Bhandari et al. (2018), Section 8.2. Adding projection results in Algorithm 4.

Guarantees of convergence of Algorithm 4 are given in Theorem 5.

**Theorem 5.** *Suppose Assumptions 1, 2, 4 hold, then output of Algorithm 4 will satisfy:*

$$E[f(\tilde{\theta}_s)] \leqslant (\frac{3}{4})^s f(\theta_0) + \frac{8C}{\lambda_{\min}n_m} + 4\eta(2G^2(4 + 6\tau^{mix}(\eta)) + 9R^2),$$

*where $C = \frac{4(1+(m-1)\rho)}{(1-\rho)}[4R^2 + r_{\max}^2]$.*

*Proof.* The proof is given in Appendix G.1.  □

Theorem 5 implies that if we choose $s = \mathcal{O}(\log(1/\epsilon))$, $n_m = \mathcal{O}(1/(\lambda_{\min}\epsilon))$ and $\eta = \mathcal{O}(\epsilon/\log(1/\epsilon))$ and $M = \mathcal{O}(\frac{\log(1/\epsilon)}{\epsilon\lambda_{\min}})$, total sample complexity to achieve accuracy of $\epsilon$ is:

---

**Algorithm 4** TD-SVRG with batching for Markovian sampling case

---

**Parameters** update frequency $M$, learning rate $\eta$, projection radius $R$ and batch size $n_m$

**Initialize** $\tilde{\theta}_0$.

**Iterate:** for $m = 1, 2, \ldots$
    $\theta = \tilde{\theta}_{m-1}$,
    sample trajectory $\mathcal{D}^m$ of length $n_m$,
    compute $\mu = \frac{1}{n_m} \sum_{s,s' \in \mathcal{D}^m} g_{s,s'}(\theta)$,
    where $g_{s,s'}(\theta) = (r(s,s') + \gamma\phi(s')^T\theta - \phi(s)^T\theta)\phi(s_t)$,
    $\theta_0 = \tilde{\theta}$.
    **Iterate:** for $t = 1, 2, \ldots, M$
        Randomly sample $s, s'$ and compute update vector
        $v_t = g_{s,s'}(\theta_{t-1}) - g_{s,s'}(\tilde{\theta}) + \mu$,
        Update parameters $\theta_t = \Pi_R(\theta_{t-1} - \eta v_t)$.
    **end**
    set $\tilde{\theta}_m = \theta_t$ for randomly chosen $t \in (0, \ldots, M-1)$.
**end**

---

$$\mathcal{O}(\frac{\log^2(1/\epsilon)}{\epsilon\lambda_{\min}})$$

In the most practical application his result is better than $\mathcal{O}(\frac{1}{\epsilon\lambda_{\min}^2}\log(1/\epsilon))$, since $\log(1/\epsilon)/\lambda_{\min} > 1$ for practical values of $\epsilon$ and $\lambda_{\min}$.

### G.1 PROOF OF THEOREM 5

In the Markovian sampling case, we cannot simply apply Lemma 1; due to high estimation bias the bounds on $f(\theta)$ and $w(\theta)$ will not be derived based on current value of $\theta$, but based on global constraints on the updates guaranteed by applying projection.

First, we analyse a single iteration on step $t$ of epoch $m$, during which we apply the update vector $v_t = g_t(\theta) - g_t(\tilde{\theta}) + \mu(\tilde{\theta})$. The update takes the form:

$$
\begin{aligned}
E\|\theta_t - \theta^*\|_2^2 = E\|\Pi_R(\theta_{t-1} + \eta v_t) - \Pi_R(\theta^*)\|_2^2 &\leqslant E\|\theta_{t-1} - \theta^* + (-\eta v_t)\|_2^2 = \\
&\|\theta_{t-1} - \theta^*\|_2^2 + 2\eta(\theta_{t-1} - \theta^*)^T E[v_t] + \eta^2 E\|v_t\|_2^2 = \\
&\|\theta_{t-1} - \theta^*\|_2^2 + 2\eta(\theta_{t-1} - \theta^*)^T(\mathbb{E}[g_t(\theta_{t-1})] - \mathbb{E}[g_t(\tilde{\theta})] + \mu(\tilde{\theta})) + \\
&\eta^2 E\|v_t\|_2^2,
\end{aligned} \tag{6}
$$

where the expectation is taken with respect to $s, s'$ sampled during iteration $t$. Recall that under Markovian sampling, $\mathbb{E}[g_t(\theta_{t-1})] \neq \bar{g}(\theta_{t-1})$ and that for the expectation of the estimated mean-path update we have $\mathbb{E}[\mu(\tilde{\theta})|s_{m-1}] \neq \bar{g}(\tilde{\theta})$, where $s_{m-1}$ is the last state of epoch $m-1$. To tackle this issue, we follow the approach introduced in a previous works (Bhandari et al. (2018), Xu et al. (2020)) and rewrite the expectation as a sum of mean-path update and error terms. Similar to Bhandari et al. (2018), we denote the error term on a single update as $\zeta$:

$$\zeta_t(\theta) = (\theta - \theta^*)^T(g_t(\theta) - \bar{g}(\theta)).$$

For an error term on the trajectory we follow Xu et al. (2020) and denote it as $\xi$:

$$\xi_m(\theta) = (\theta - \theta^*)^T(\mu(\theta) - \bar{g}(\theta)).$$

Applying this notation, 6 can be rewritten as:

$$
\begin{aligned}
E\|\theta_t - \theta^*\|_2^2 \leqslant &\|\theta_{t-1} - \theta^*\|_2^2 + \\
&2\eta(\theta_{t-1} - \theta^*)^T(\mathbb{E}[g_{t-1}(\theta_{t-1})] - \mathbb{E}[g_t(\tilde{\theta})] + \mu(\tilde{\theta})) + \eta^2 E\|v_t\|_2^2 = \\
&\|\theta_{t-1} - \theta^*\|_2^2 + 2\eta\big[(E[\zeta_t(\theta_{t-1})] + (\theta_{t-1} - \theta^*)^T\bar{g}(\theta_{t-1})) - \\
&(E[\zeta_t(\tilde{\theta})] - (\theta_{t-1} - \theta^*)^T\bar{g}(\tilde{\theta})) + \\
&(E[\xi(\tilde{\theta})] - (\theta_{t-1} - \theta^*)^T\bar{g}(\tilde{\theta}))\big] + \eta^2 E\|v_t\|_2^2.
\end{aligned} \tag{7}
$$

Error terms can be bounded by slightly modified lemmas from the original papers. For $\zeta(\theta)$, we apply a bound from Lemma 11 in Bhandari et al. (2018):

$$|E[\zeta_t(\theta)]| \leqslant G^2(4 + 6\tau^{mix}(\eta))\eta. \tag{8}$$

In the original lemma, a bound on $E[\zeta_t(\theta)]$ is stated, however, in the proof a bound on absolute value of the expectation is also derived.

For mean-path estimation error term, we use a modified version of Lemma 1 in Xu et al. (2020). The proof of this lemma in the original paper starts by applying the inequality

$$a^T b \leqslant \frac{k}{2}||a||^2 + \frac{1}{2k}||b||^2$$

to the expression $(\theta - \theta^*)^T(\mu(\theta) - \bar{g}(\theta))$, with $k = \lambda_A/2$ (using the notation in Xu et al. (2020)). For the purposes of our proof we use $k = \lambda_{\min}$. Thus, we will have the expression:

$$\begin{aligned}
\mathbb{E}[\xi_m(\theta)] \leqslant &\frac{\lambda_{\min}}{2}\mathbb{E}[||\theta - \theta^*||_2^2|s_{m-1}] + \frac{4(1 + (m-1)\rho)}{\lambda_{\min}(1-\rho)n_m}[4R^2 + r_{\max}^2] = \\
&\frac{\lambda_{\min}}{2}\mathbb{E}[||\theta - \theta^*||_2^2|s_{m-1}] + \frac{C}{\lambda_{\min}n_m}.
\end{aligned} \tag{9}$$

Also, note, that the term $E||v_t||_2^2$ might be bounded as $E||v_t||_2^2 \leqslant 18R^2$. Plugging 8 and 9 bounds into 7 we obtain:

$$\begin{aligned}
E||\theta_t - \theta^*||_2^2 \leqslant &||\theta_{t-1} - \theta^*||_2^2 - 2\eta f(\theta_{t-1}) + 4\eta^2 G^2(4 + 6\tau^{mix}(\eta)) + \\
&2\eta(\frac{\lambda_{\min}}{2}||\tilde{\theta} - \theta^*||_2^2 + \frac{C}{\lambda_{\min}n_m}) + 18\eta^2 R^2.
\end{aligned}$$

Summing the inequality over the epoch and taking expectation with respect to all previous history, we have:

$$\begin{aligned}
2\eta M E[f(\tilde{\theta}_s)] \leqslant &||\tilde{\theta}_{s-1} - \theta^*||_2^2 + 2\eta M(\frac{\lambda_{\min}}{2}||\tilde{\theta}_{s-1} - \theta^*||_2^2 + \frac{C}{\lambda_{\min}n_m}) + \\
&\eta^2 M(4G^2(4 + 6\tau^{mix}(\eta)) + 18R^2).
\end{aligned}$$

Then we divide both sides by $2\eta M$ and use $||\tilde{\theta}_{s-1} - \theta^*||_2^2 \leqslant 1/\lambda_{\min}f(\tilde{\theta}_{s-1})$ to obtain:

$$\begin{aligned}
E[f(\tilde{\theta}_s)] \leqslant &(\frac{1}{2\lambda_{\min}\eta M} + \frac{1}{2})f(\tilde{\theta}_{s-1}) + \frac{C}{\lambda_{\min}n_m} + \\
&\eta(2G^2(4 + 6\tau^{mix}(\eta)) + 9R^2).
\end{aligned}$$

We choose $\eta$ and $M$ such that $\eta M \lambda_{\min} = 2$. We then apply this inequality to the value of the function $f$ in the first term in the right-hand side recursively, which yields the desired result:

$$E[f(\tilde{\theta}_s)] \leqslant (\frac{3}{4})^s f(\theta_0) + \frac{8C_0}{\lambda_{\min}n_m} + 4\eta(2G^2(4 + 6\tau^{mix}(\eta)) + 9R^2)$$

# H   DETAILS ON ALGORITHM IMPLEMENTATION

## H.1   COMPARISON OF THEORETIC BATCH SIZES

In this subsection we compare the values of batch sizes which are theoretically required to guarantee convergence. We compare batch sizes of three algorithms: TD-SVRG, PDSVRG (Du et al. (2017)) and VRTD (Xu et al. (2020)). Note that PDSVRG and VRTD are algorithms for different settings, but for TD-SVRG the batch size value is the same: $16/\lambda_{\min}$, thus, we compare two algorithms in the same table. We compare the batch sizes required by algorithm for three MDPs: first with 50 state, 20 action and $\gamma = 0.8$, second with 400 state, 10 actions and $\gamma = 0.95$, third with 1000 states, 20 actions and $\gamma = 0.99$, with actions choice probabilities generated from $U[0, 1]$. (similar to one used for experiments in Subsections 7.1 and 7.2). Since a batch size is dependent on the smallest eigenvalue of the matrix $A$, which is, in turn, is dependent on the dimensionality of the feature vector, we do the comparison for different feature vector sizes: 5, 10, 20 and 40 randomly

generated features + 1 constant feature for each state. We generate 10 datasets and environments for each feature size. Our results are summarized in tables 2, 3 and 4

Table 2: Comparison of theory suggested batch sizes for MDP with 50 states, 20 actions and $\gamma = 0.8$. Values in the first row are feature vectors dimensionality. Value in other rows: bitch size of a corresponded method (row 1). Values are average over 10 generated datasets and environments.

| Method/Features | 6 | 11 | 21 | 41 |
|---|---|---|---|---|
| TD-SVRG | 2339 | 6808 | 21553 | $4.51 \cdot 10^5$ |
| PD SVRG | $1.52 \cdot 10^{16}$ | $3.09 \cdot 10^{19}$ | $1.85 \cdot 10^{23}$ | $1.41 \cdot 10^{36}$ |
| VRTD | $3.07 \cdot 10^6$ | $2.13 \cdot 10^7$ | $3.79 \cdot 10^8$ | $165 \cdot 10^{11}$ |

Table 3: Comparison of theory suggested batch sizes for MDP with 400 states, 10 actions and $\gamma = 0.95$. Values in the first row are feature vectors dimensionality. Value in other rows: bitch size of a corresponded method (row 1). Values are average over 10 generated datasets and environments.

| Method/Features | 6 | 11 | 21 | 41 |
|---|---|---|---|---|
| TD-SVRG | 3176 | 6942 | 18100 | 54688 |
| PD SVRG | $1.72 \cdot 10^{16}$ | $3.83 \cdot 10^{18}$ | $3.06 \cdot 10^{21}$ | $5.77 \cdot 10^{24}$ |
| VRTD | $5.41 \cdot 10^6$ | $2.53 \cdot 10^7$ | $1.63 \cdot 10^8$ | $1.58 \cdot 10^9$ |

Table 4: Comparison of theory suggested batch sizes for MDP with 1000 states, 20 actions and $\gamma = 0.99$. Values in the first row are feature vectors dimensionality. Value in other rows: bitch size of a corresponded method (row 1). Values are average over 10 generated datasets and environments.

| Method/Features | 6 | 11 | 21 | 41 |
|---|---|---|---|---|
| TD-SVRG | 9206 | 16096 | 32723 | 79401 |
| PD SVRG | $7.38 \cdot 10^{18}$ | $9.64 \cdot 10^{20}$ | $5.14 \cdot 10^{23}$ | $4.97 \cdot 10^{26}$ |
| VRTD | $4.35 \cdot 10^7$ | $1.34 \cdot 10^8$ | $5.44 \cdot 10^8$ | $1.45 \cdot 10^9$ |

## H.2 BATCHED SVRG PERFORMANCE

In this set of experiments we compare the performance of TD-SVRG and batched TD-SVRG in finite-sample case. We generate 10 datasets of size 50000 from the similar MDP as in Section 7.1. Algorithms also run with the same hyperparameters. Average results over 10 runs presented on Figure 3 and show, that batched TD-SVRG saves a lot of computations during the earlier epochs, which provides faster convergence.

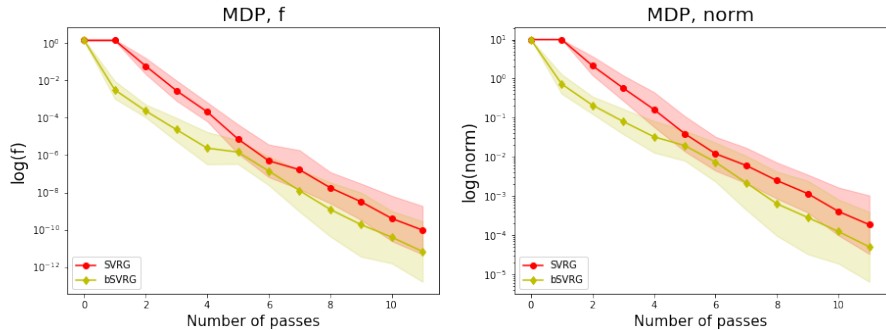

Figure 3: Average performance of TD-SVRG and batching TD-SVRG in finite sample case. Datasets sampled from MDP environments. Left figure - performance in terms of $\log(f(\theta))$. Right figure - performance in terms of $\log(|\theta - \theta^*|)$.

