# OpenReview forum: "Closing the Gap Between SVRG and TD-SVRG with Gradient Splitting"
_ICLR.cc/2023/Conference — Submitted to ICLR 2023_

### Official Review · Reviewer_kaRJ · 2022-10-15

**Confidence:** 3
**Correctness:** 4
**Technical Novelty And Significance:** 2
**Empirical Novelty And Significance:** Not applicable
**Recommendation:** 5

**Clarity, Quality, Novelty And Reproducibility:**

The authors developed the key lemma 1, which allows them to establish an analysis analogous to the original analysis of SVRG in the convex setting. This simpler analysis is based on introducing the functions $f,\omega$, and provides a better understanding of TD under variance reduction. I suggest the authors clarify the novelty of this lemma, i.e., how does this lemma differ from the existing works that derive similar type of bounds?

**Strength And Weaknesses:**

Strength:

-The presentation is clear and easy to follow

-This paper is the first to prove that TD with SVRG can match the complexity result of SVRG in convex optimization.

Weakness:

-Paper writing is poor. There are many grammar issues, incomplete sentences, etc.


**Summary Of The Paper:**

This paper develops a reanalysis of TD with SVRG variance reduction and establishes a tight bound that matches the complexity result of SVRG in convex optimization. The developed analysis is simpler than all previous research and leads to better convergence bounds. Moreover, numerical experiments show the advantage of this algorithm over other existing variance-reduced TD algorithms.

**Summary Of The Review:**

See above.

---

> ### Author Response · Authors · 2022-11-19
> **Answer to review by Reviewer kaRJ**
>
> Thank you for your comments and suggestions!
>
> **-Paper writing is poor. There are many grammar issues, incomplete sentences, etc.**
>
> We apologize for the rough state of the initial submission. We made an effort to eliminate grammatical errors and fix typos in the updated submission. While English is not the native language of any of the authors, we believe the paper is understandable
>
> **Clarity: The authors developed the key lemma 1, which allows them to establish an analysis analogous to the original analysis of SVRG in the convex setting. This simpler analysis is based on introducing the functions $f$ and $w$ , and provides a better understanding of TD under variance reduction. I suggest the authors clarify the novelty of this lemma, i.e., how does this lemma differ from the existing works that derive similar type of bounds?**
>
> The key idea in our work is the idea of a gradient splitting. In brief, a key step in the analysis of any optimization method $\theta_{t+1} = \theta_t + \alpha_t \Delta_t$ is the analysis of the quantity
>
> $$ E [ \Delta_t^T (\theta_t - \theta^*)], $$ i.e., the inner product between the direction of motion and the direction to the optimal solution. Using gradient splitting tells us exactly what it is -- it is exactly the same as the product
>
> $$ E [ \nabla F(\theta_t)^T (\theta_t - \theta^*),$$ for a function $F(\theta)$ that we can write out explicitly:
>
> $$ F(\theta) = (1-\gamma) D(V_{\theta} - V_{\theta^*})^2 + \gamma Dir(V_{\theta} - V_{\theta^*})^2, $$with the definitions of the D and Dir norms given in the paper.
>
> Now this idea allows us to bring all sorts of ideas from optimization into RL. In particular, we can use this to follow the original SVRG proof, with Lemma 1 being the key step -- and after Lemma 1, the proof is reduced to establishing bounds on $f$, $w$ and $||\theta - \theta^* ||^2$ in terms of the target function. We have modified the discussion after Lemma 1 to clarify this point.
>
> By contrast, previous literature lacked the direct analogy between gradient descent and TD that we use here, and had to rely on much more involved analyses and often more involved (primal-dual) methods. Frankly, we find many of the previous papers difficult to follow due to the intricacies of the analysis. Our use of the gradient splitting ideas allows us to derive Lemma 1 directly, and quite easily -- our proofs are very short! -- without needing the more sophisticated and involved analyses from the previous work.

---

### Official Review · Reviewer_Tp1f · 2022-10-20

**Confidence:** 4
**Correctness:** 4
**Technical Novelty And Significance:** 1
**Empirical Novelty And Significance:** Not applicable
**Recommendation:** 1

**Clarity, Quality, Novelty And Reproducibility:**

The quality of writting should be further improved.

The clarity is poor. Many necessary concepts are not introduced in this paper. How are the finite-sample and iid online sampling senario related to MDP? How do you split the gradient and what is the gradient for TD-learning algorithm?

No originality. Applying the variance reduction technique to TD-learning is not new. The proof technique is not new neither. The sample complexity improvement is not satisfactory (unless the theoretical lower bound is provided).

**Strength And Weaknesses:**

***Strength***
This paper improves the sample complexity of an existing convergence result of the VRTD algorithm.

***Weaknesses***
1. In RL, online learning cannot have iid sampling. It can be a good starting point but the author should consider the Markovian sampling structure as [Xu2019].

[Xu2019] Xu T, Wang Z, Zhou Y, Liang Y. Reanalysis of Variance Reduced Temporal Difference Learning. In International Conference on Learning Representations 2019 Sep 25.

2. The theoretical result only obtains a constant-level complexity improvement. I don't think it fills any gap unless the theoretical lower bound is also provided; TD-learning is not equivalent to SGD in convex optimization. Also, there exists theoretical results showing that variance-reduction technique can make a variant of TD-learning algorithm achieve better complexity than $\mathcal{O}(\epsilon^{-1}\log \epsilon^{-1})$, see [Ma2020].

[Ma2020] Ma S, Zhou Y, Zou S. Variance-reduced off-policy TDC learning: Non-asymptotic convergence analysis. Advances in Neural Information Processing Systems. 2020;33:14796-806.

3. The gradient splitting method seems unrelated to this paper, even if it is put in the title. To my understanding, it doesn't bring any high-level perspectives to make the reader have a new understanding on TD-learning algorithm; it is just a trick used to derive the bound. Every optimization paper can have a lot of such tricks.


**Summary Of The Paper:**

This paper provides a theoretical convergence proof for TD-learning algorithm with variance reduction structure. The sample complexity can match the complexity of SGD for convex optimization in iid finite-sample and iid online sampling senario. The empirical study furthor verifies the performance improvement from the variance reduction technique.

**Summary Of The Review:**

In summary, I will reject this paper because: (1) both algorithm and proof technique are not new; (2) the problem setting is not possible in RL.

---

> ### Author Response · Authors · 2022-11-19
> **Answer to review by by Reviewer Tp1f**
>
> Thank you for your comments and suggestions.
>
> **1. In RL, online learning cannot have iid sampling. It can be a good starting point but the author should consider the Markovian sampling structure as [Xu2019].**
>
> Following the reviewer's suggestion, we added the analysis of the Markovian sampling to the paper (Appendix Section G). The convergence bound we achieve is $\mathcal{O}(\frac{\log^2(1/\epsilon)}{\epsilon \lambda_{min}})$, which is better than results obtained by [Xu2019]  $\mathcal{O}(\frac{\log(1/\epsilon)}{\epsilon \lambda^2_{min}})$in the most practical problems.
>
> **2. The theoretical result only obtains a constant-level complexity improvement. I don't think it fills any gap unless the theoretical lower bound is also provided; TD-learning is not equivalent to SGD in convex optimization.**
>
> We would submit to the reviewer that it is incorrect to refer to our results as a constant-level complexity improvement. Indeed, we are improving the scaling *with the condition number of the problem*, by reducing it from squared to linear. If one considers this as a constant-level complexity improvement, then Nesterov acceleration is also a constant-level complexity improvement. So is the SVRG itself, which, like this paper, shows how to minimize the sum of functions with a better dependence on the condition number.
>
> More broadly, by considering Nesterov acceleration and the SVRG, it should be clear that improved scaling with condition numbers can be a big deal! To truly appreciate the magnitude of this improvement, here are a few tables showing how the improved scalings translate into recommended batch sizes (added to the paper, Appendix Subsection H.1):
>
> 50 states:
> | Method\Features|6|11|21|41|
> |-|-|-|-|-|
> | TD-SVRG  |  $2339$ | $6808$ |  $21553$| $4.51\cdot10^5$|
> | PD SVRG  |    $1.52\cdot10^{16}$ | $3.09\cdot10^{19}$ |  $1.85\cdot 10^{23}$ | $1.41\cdot10^{36}$|
> |  VRTD |   $3.07\cdot10^{6}$ | $2.13\cdot10^{7}$ | $3.79\cdot10^{8}$ | $165\cdot10^{11}$   |
>
> 400 states:
> | Method\Features  |  6 | 11  | 21  | 41  |
> |-|-|-|-|-|
> | TD-SVRG  |  $3176$ | $6942$ |  $18100$ | $54688$ |
> | PD SVRG  |    $1.72\cdot10^{16}$ | $3.83\cdot10^{18}$ |  $3.06\cdot 10^{21}$ | $5.77\cdot10^{24}$|
> |  VRTD |   $5.413\cdot10^{6}$ | $2.53\cdot10^{7}$ | $1.63\cdot10^{8}$ | $1.58\cdot10^9$   |
>
> 1000 states:
> | Method\Features  |  6 | 11  | 21  | 41  |
> |-|-|-|-|-|
> | TD-SVRG  |  $9206$ | $16096$ | $32723$| $79401$|
> | PD SVRG  |   $7.38\cdot10^{18}$ | $9.64\cdot10^{20}$|  $5.14\cdot 10^{23}$ | $4.97\cdot10^{26}$|
> |  VRTD |   $4.35\cdot10^{7}$ | $1.34\cdot10^{8}$ | $5.44\cdot10^{8}$ | $1.45\cdot10^9$  |
>
> These tables show that condition numbers are important in TD problems and this is the reason why many authors consider them in their analysis, including [Xu2019], [Bhandari2018],[Du2017], [Korda2015], [Peng2019] and others.
>
> Finally, asking for a lower bound in this context sets an impossible task before us: SVRG does not match any lower bounds, so its modification for the TD setting is unlikely to do so.
>
>
> **Also, there exists theoretical results showing that variance-reduction technique can make a variant of TD-learning algorithm achieve better complexity than $\mathbb{O} (\epsilon^{-1} \log \epsilon^{-1})$, see [Ma2020].**
>
> We want to thank the reviewer for pointing out the reference [Ma et al., 2020], which was unfamiliar to us. We will look at it carefully. For now, however, we want to remark that in our setting, it is not possible to improve on $O(\epsilon^{-1})$ sample complexity, for reasons we explain next. We will examine [Ma et al, 2020] to see how their assumptions differ and insert a reference into the final result; however, our core response is that whatever assumptions they are making to achieve the improvement make their result inapplicable to our setting.
>
> The core reason why better than $O(\epsilon^{-1})$ sample complexity is impossible is that policy evaluation contains, as a subproblem, the problem of estimating a mean of an unknown random variable. Indeed, consider an MDP with one state and one action. The reward on the action is a random variable which we call $X$. Let us assume that the discount factor is $1/2$. In that case, the value function is simple $E[2X]$. Thus policy evaluation is nothing more than mean estimation for this MDP.
>
> Now how well can you estimate mean from $t$ samples? It is not hard to see that the best thing to do, given samples $X_1, \ldots, X_t$, is to average them $$ \bar{X}_t = \frac{X_1 + \cdots + X_t}{t}.$$ The variance here will be on the order of $\sim 1/t$. Thus the time until the expectation of the squared error is less than $\epsilon$ is going to be $O(\epsilon^{-1})$.
>
> **Regarding the term "closing the gap", we refer to the similarity in obtained convergence rates in finite sample and iid cases with SVRG in the convex optimization case. %These two cases are taking because update sampling strategy is similar to classical SVRG.**
>
> This fact is discussed in Section 4.4.

---

> ### Author Response · Authors · 2022-11-19
> **Answer to review by Reviewer Tp1f (question 3)**
>
> **3. The gradient splitting method seems unrelated to this paper, even if it is put in the title. To my understanding, it doesn't bring any high-level perspectives to make the reader have a new understanding on TD-learning algorithm; it is just a trick used to derive the bound. Every optimization paper can have a lot of such tricks.**
>
> While one could say that, but then it is the only trick we have used which was not present in the earlier literature. It is thus this trick which is responsible for the improvements we report.
>
> More generally, we argue that the splitting idea can bring a new high-level perspective into the analysis of TD learning. Indeed, a key step in the analysis of any optimization method $\theta_{t+1} = \theta_t + \alpha_t \Delta_t$ is the analysis of the quantity
>
> $$ E [ \Delta_t^T (\theta_t - \theta^*)], $$
>  i.e., the inner product between the direction of motion and the direction to the optimal solution. Using gradient splitting tells us exactly what it is -- it is exactly the same as the product
>
> $$ E [ \nabla F(\theta_t)^T (\theta_t - \theta^*)], $$
>
>  for a function $F(\theta)$ that we can write out explicitly:
>
> $$ F(\theta) = (1-\gamma) D(V_{\theta}- V_{\theta^*} )^{2} + \gamma Dir(V_{\theta} - V_{\theta}^*)^2, $$
>
>  with the definitions of the D and Dir norms given in the paper. This allows all sorts of ideas from gradient descent to be used in RL, and the current paper is merely one application of this.

---

### Official Review · Reviewer_pncF · 2022-10-25

**Confidence:** 3
**Correctness:** 3
**Technical Novelty And Significance:** 2
**Empirical Novelty And Significance:** 2
**Recommendation:** 5

**Clarity, Quality, Novelty And Reproducibility:**

Novelty: This paper applied the well-known SVRG technique to the TD method to achieve faster convergence. It is new but seems not very interesting.

Quality: The theoretical analysis is technically sound with excellent analysis. Experimental validation is provided.

Clarity: Some concepts in the paper is unclear. What is 'Gradient Splitting' in the title?

Reproducibility: The code needed to reproduce the experimental results is not provided.



**Strength And Weaknesses:**

Strengths:

1: This paper proposes a variant of the TD method with a faster convergence by introducing the well-known SVRG technique.

2: Extensive experiments are conducted to verify the effectiveness and efficiency of the proposed algorithm.


Weaknesses:

1: It is unclear whether the proposed method significantly improves the existing complexity in VRDT. For the overall complexity, which term will dominate the maximization function in theory or in practice? Since the superiority of the proposed method to the VRDT method depends on this, the authors would do well to provide more analysis.

**Summary Of The Paper:**

This paper studies the Temporal difference (TD) learning method for policy evaluation in reinforcement learning. The proposed approach TD-SVRG method is a variant of the TD method by introducing the well-known SVRG technique. Theoretically, their analysis can lead to better convergence bounds for previous methods. Numerical results validate the improved performance of the proposed method over the existing methods.

**Summary Of The Review:**

This paper proposes a variant of the TD method with a faster convergence by introducing the well-known SVRG technique. The novelty and the improvements are limited.

------Update------

I have read the author's response and would like to keep the score.

---

> ### Author Response · Authors · 2022-11-18
> **Answer to review by Reviewer pncF**
>
> Thank you for your comments and suggestion!
>
> **Weaknesses 1: It is unclear whether the proposed method significantly improves the existing complexity in VRDT. For the overall complexity, which term will dominate the maximization function in theory or in practice? Since the superiority of the proposed method to the VRDT method depends on this, the authors would do well to provide more analysis.**
>
> We provide a comparison below. Our key point is that the quantity $\lambda_{A}^{-2}$ can be astronomically large on even very simple examples. Thus reducing the scaling from $\lambda_{A}^{-2}$ to $\lambda_{A}^{-1}$ will dominate the performance by many orders of magnitude.
>
> To illustrate this, we have added Subsection H.1, in which proposed values are compared in for various dimensionalities for feature vector, containing the following tables:
>
> 50 states:
> | Method\Features  |  6 | 11  | 21  | 41  |
> |---|---|---|---|---|
> | TD-SVRG  |  $2339$ | $6808$ |  $21553$| $4.51\cdot10^5|
> | PD SVRG  |    $1.52\cdot10^{16}$ | $3.09\cdot10^{19}$ |  $1.85\cdot 10^{23}$ | $1.41\cdot10^{36}$|
> |  VRTD |   $3.07\cdot10^{6}$ | $2.13\cdot10^{7}$ | $3.79\cdot10^{8}$ | $165\cdot10^{11}$   |
>
> 400 states:
> | Method\Features  |  6 | 11  | 21  | 41  |
> |---|---|---|---|---|
> | TD-SVRG  |  $3176$ | $6942$ |  $18100$ | $54688$ |
> | PD SVRG  |    $1.72\cdot10^{16}$ | $3.83\cdot10^{18}$ |  $3.06\cdot 10^{21}$ | $5.77\cdot10^{24}$|
> |  VRTD |   $5.413\cdot10^{6}$ | $2.53\cdot10^{7}$ | $1.63\cdot10^{8}$ | $1.58\cdot10^9$   |
>
> 1000 states:
> | Method\Features  |  6 | 11  | 21  | 41  |
> |---|---|---|---|---|
> | TD-SVRG  |  $9206$ | $16096$ | $32723$| $79401$|
> | PD SVRG  |   $7.38\cdot10^{18}$ | $9.64\cdot10^{20}$|  $5.14\cdot 10^{23}$ | $4.97\cdot10^{26}$|
> |  VRTD |   $4.35\cdot10^{7}$ | $1.34\cdot10^{8}$ | $5.44\cdot10^{8}$ | $1.45\cdot10^9$  |
>
> **Clarity: Some concepts in the paper is unclear. What is 'Gradient Splitting' in the title?**
>
> The term "Gradient Splitting" is taken from "Temporal Difference Learning as Gradient
> Splitting” [Rui 2020]: linear function $h(x) = B(x-a)$ is **gradient splitting** of quadratic function $j(x) = (x-a)^T Q(x-a)$, where $Q$ is symmetric positive semi-definite matrix, if $B+B^T = 2Q$. We give the formal definition of this term in the paper, Subsection 2.1, we modified the term to be in bold text to make it more visible.
>
> **Reproducibility: The code needed to reproduce the experimental results is not provided.**
>
> Authors included a [link](https://anonymous.4open.science/r/SVRG_for_TD_learning-C688) to anonymous repository to the paper so that now experimental results can be reproduced.

---

### Official Review · Reviewer_158U · 2022-10-25

**Confidence:** 3
**Correctness:** 3
**Technical Novelty And Significance:** 4
**Empirical Novelty And Significance:** 3
**Recommendation:** 6

**Clarity, Quality, Novelty And Reproducibility:**

**Clarity:**
The paper is very clear conceptually, but there are quite a few grammatical mistakes that make it harder to read, including missing words and typographic errors. The discussion of results is very clear, and helpful explanations are provided throughout.

**Quality:**
I have some concerns about the experiments, detailed in the Strengths/Weaknesses section of this review.

**Novelty:**
To the best of my knowledge, the analysis is novel.

**Reproducibility:**
The paper is missing a few details that would be helpful for reproducing its experiments.

**Strength And Weaknesses:**

**Strengths:**
1. The analysis uses the gradient splitting approach, which yields a significantly simpler algorithm, analysis, learning rate/step size, and a better convergence rate.
1. The paper considers several different settings, which increases the number of readers who would be interested.
1. The paper provides recommendations for setting the learning rate, and actually uses this recommended learning rate in the experiments.

**Potential Weaknesses/Questions:**
1. Many grammatical errors, including typographic errors ("TD-leaning", "apprxoimate") and missing words, especially definite and indefinite articles. For example, "determining *the* expected reward *an* agent will achieve if it chooses *actions* according to *a* stationary policy". Some grammatical issues are listed below, but there were too many for me to write down all of them.
1. ~~The paper doesn't comment on the assumptions made other than stating that they are standard. It would be nice to include a sentence or two explaining why the assumptions are reasonable or whether they are just for convenience (like Assumption 2; the feature vectors can always be normalized to make it true).~~
1. Jumping between the notation used in other papers and the notation used in this paper is difficult. It would be better to convert the results of other papers to the notation used in this paper.
1. ~~Font size of figures is too small to read without zooming in a lot.~~
1. I have some concerns about the experiments:
    1. ~~Why weren't the parameters for each algorithm set using a grid search the way they were for PD-SVRG? Without doing this, it's not clear that the chosen parameters are representative of the performance of each algorithm, and hence no conclusion about the performance of each algorithm can be drawn.~~
    1. ~~Why was the learning rate for TD set to $1/\sqrt{t}$? It seems like the algorithm stops learning very quickly due to the learning rate shrinking so quickly. How does the algorithm perform with a learning rate of $1/t$ instead?~~ I realized I was confused about this.
    1. ~~What were the parameter values checked during the grid search? This would make the experiments more reproducible.~~
    1. ~~Either confidence intervals or standard error (itself a form of confidence interval, I guess) should be included in the plots to communicate statistical significance to the reader.~~
    1. Consider using either colourblind-friendly colours in the plots, or replacing the legend with labels for each line to remove the dependence on colour to determine which algorithm is which. Using differently-shaped points for each algorithm is a good start, but when the points are too close together it becomes difficult to tell which algorithm is which.
    1. ~~What is the threshold for removing highly correlated features? This would help reproducibility.~~
1. ~~The paper states that TD-SVRG and PD-SVRG converge linearly, but the y-axis of the plots appears to be in log space, which is confusing. Could this be clarified?~~
1. ~~It would be good to have a concluding section that summarizes the main takeaways of the paper.~~

**Grammatical/typographic issues:**
1. "Robbins & Munro (1951)" should have the authors names inside the parentheses, because they are not being referred to in the sentence. Also, I think the second author's name is spelled "Monro".
1. The "Korda & LA (2014)" citation is wrong: the authors are Nathaniel Korda and L.A. Prashanth, and the paper was published at ICML 2015.
1. "These methods are collectively known as variance reduction." Should this be, "These methods are collectively known as variance-reduced gradient methods."?
1. "We analyze this case in 4 and 5": It's better to write "Sections 4 and 5", because otherwise "4 and 5" could refer to an equation, appendix, theorem, etc.
1. In Algorithm 1, should N be M? If not, it would be good to define N explicitly.
1. "unbalacedness"
1. "is the the size"
1. "on practice" should be "in practice"
1. "envtironments" in Figure 1.

**Summary Of The Paper:**

The paper uses SVRG to reduce variance in TD learning. The resulting algorithm is analyzed using the gradient splitting perspective on TD learning to prove finite sample convergence rates that match those of SVRG in the setting of convex optimization. The analysis is done for several settings, and the findings are illustrated with experiments on several environments.

**Summary Of The Review:**

~~Despite really liking this paper, I must recommend the current version be rejected due to concerns about clarity and the experiments detailed above. If the authors address these concerns satisfactorily and no critical issues are found by the other reviewers, I would recommend acceptance of a revised version.~~ The authors have addressed most of my concerns in their response, except some of my concerns about clarity (making these clarity changes would drastically reduce the work required of each reader to understand the paper). I have increased my score to reflect this.

---

> ### Author Response · Authors · 2022-11-18
> **Answer to review by Reviewer 158U**
>
> Thank you for your comments and suggestions! We address your question (except 5) and concerns in the same order:
>
> **1. Many grammatical errors, including typographic errors ("TD-leaning", "apprxoimate") and missing words, especially definite and indefinite articles. For example, "determining the expected reward an agent will achieve if it chooses actions according to a stationary policy". Some grammatical issues are listed below, but there were too many for me to write down all of them.**
>
> We apologize for the inconvenience we have caused. We made an best effort to eliminate grammatical errors and fix typos in the updated submission. Note that none of the authors' native language has definite/indefinite articles. Nevertheless, despite the grammatical errors, we believe our current draft is understandable.
>
> **2. The paper doesn't comment on the assumptions made other than stating that they are standard. It would be nice to include a sentence or two explaining why the assumptions are reasonable or whether they are just for convenience (like Assumption 2; the feature vectors can always be normalized to make it true).**
>
> Following the reviewer's suggestion we added comments right after Assumptions 1 and 2.
>
> **3. Jumping between the notation used in other papers and the notation used in this paper is difficult. It would be better to convert the results of other papers to the notation used in this paper.**
>
> We will address the notational issues in the revised version. In general, we employ notation similar to finite-sample literature, as in Du (2017) and Peng (2019).
>
> Regarding Table 1, we follow the notation introduced by the authors of the corresponding papers, without properly defining every symbol for space-saving reasons.
>
> **4. Font size of figures is too small to read without zooming in a lot.**
>
> Following the reviewer's  suggestion we increased the font size of the figures.
>
> **6. The paper states that TD-SVRG and PD-SVRG converge linearly, but the y-axis of the plots appears to be in log space, which is confusing. Could this be clarified?**
>
> While writing the paper, we used  "linear convergence" and "geometric convergence" interchangeably (there are some resources that support this point, e.g. [1](https://glossary.informs.org/ver2/mpgwiki/index.php/Convergence), [2](https://planetmath.org/linearconvergence), [3](https://en.wikipedia.org/wiki/Rate_of_convergence)). Still, to avoid any possible confusion, we have replaced the term "linear convergence" with "geometric convergence" throughout the paper.
>
> **7. It would be good to have a concluding section that summarizes the main takeaways of the paper.**
>
> Following the reviewer's suggestion, the paper content was rearranged and a concluding summary was added.

---

> ### Author Response · Authors · 2022-11-18
> **Answer to Question number 5 of review by Reviewer 158U**
>
> **5. I have some concerns about the experiments:**
>
> **5.1. Why weren't the parameters for each algorithm set using a grid search the way they were for PD-SVRG? Without doing this, it's not clear that the chosen parameters are representative of the performance of each algorithm, and hence no conclusion about the performance of each algorithm can be drawn.**
>
> In our simulations we wanted to underline an important property of our algorithm: it does not require parameter grid search. For each competing algorithm we used the hyperparameters recommended by the authors of the correspondent papers. The only exception was made for PD-SVRG, because the authors of that method do not report the best set of hyperparameters. This is why we run a grid search over parameters $\sigma_w \in (1, 10^{-1}, 10^{-2})\frac{1}{\lambda_{max}(\hat{C})},\sigma_{\theta}/\sigma_w \in (10^{2},\ldots,10^{-2})$. }
>
> We adopt this strategy because it is the most robust: otherwise, one can always argue that hyperparameter search for each algorithm can be done in a different way.
>
> Incidentally, we note that our simulations show that our method, without any grid search, beats PD-SVRG with grid search. This observation further underlines the combination of theoretical and practical improvement we achieve in this paper.
>
> **5.2. Why was the learning rate for TD set to $1/\sqrt{t}$ ? It seems like the algorithm stops learning very quickly due to the learning rate shrinking so quickly. How does the algorithm perform with a learning rate of $1/t$ instead?**
>
>  We added the TD algorithm with learning rate $1/t$ to the set of experiments 1. The result in now included to Figure 1 in the updated version of the paper. This algorithm converges slower, since  $1/t \le 1\sqrt{t}, \forall t \in \mathbb{N}$.
>
> **5.3. What were the parameter values checked during the grid search? This would make the experiments more reproducible.**
>
> Code has been provided to make the experiments reproducible. We run grid search on the following parameters : $\sigma_w \in (1, 10^{-1}, 10^{-2})\frac{1}{\lambda_{max}(\hat{C})},\sigma_{\theta}/\sigma_w \in (10^{2},\ldots,10^{-2})$. Grid search was run on a separate dataset sampled from the MDP analogous to the one used in Experiment 1.
>
> **5.4. Either confidence intervals or standard error (itself a form of confidence interval, I guess) should be included in the plots to communicate statistical significance to the reader.**
>
> Following the reviewer's suggestion, we have added confidence intervals to the figures.
>
> **5.5. Consider using either colourblind-friendly colours in the plots, or replacing the legend with labels for each line to remove the dependence on colour to determine which algorithm is which. Using differently-shaped points for each algorithm is a good start, but when the points are too close together it becomes difficult to tell which algorithm is which.**
>
> To make plots more distinguishable, we decreased the total number of points, which, hopefully, made points more spread and visible.
>
> **5.6. What is the threshold for removing highly correlated features? This would help reproducibility.**
>
> We did not add discussion regarding this parameter, because it affects only dataset construction, not the algorithm performance. We have done that because highly correlated features can make solution unstable, leading to very small eigenvalues of $A$ and, consequently, very large batch sizes. We removed correlated features (correlation coefficient $>0.5$ and introduced eigen-value control (dataset was resampled if the smallest eigen-value of its matrix A was too low or too high) to avoid this issue.

---

> ### Author Response · Authors · 2022-11-30
> **Answer to updated review by Reviewer 158U**
>
> Thank you for updating you review!
>
> To further address your concerns regarding fairness of methods comparison, we run grid search for each algorithm we compare against, PD SVRG, GTD2 and VRDT. For every algorithm we tried set of parameters suggested by the authors in the Experiment section of corresponding papers. All experiments run on MDP environment with 400 states, 21 features, 10 actions and $\gamma = 0.95$, identical to one described in Section 7.1 of the paper. We run 5 experiments with $10^5$ updates for each dataset problem and 10 experiments with $3\times10^5$ for iid sampling problem, average results are compared. We added the experiment setups file to reproduce grid search and result figures to project's anonymous [github repo](https://anonymous.4open.science/r/SVRG_for_TD_learning-C688).
>
> For **PD SVRG**, we tried the exact values suggester in [Du2017], i.e. $\sigma_\theta \in (10^{-1}, \ldots, 10^{-6})\frac{1}{L_\rho \kappa (\hat{C})}$, $\sigma_w \in (1, 10^{-1}, 10^{-2}) $ and batch size is twice of the dataset size $M= 2N$. Experiment setup might be found  [here](https://anonymous.4open.science/r/SVRG_for_TD_learning-C688/exp_setup_pd_svrg_grid_search.json). Performance of 4 best performing parameter sets compared to TD SVRG might is show on this [figure](https://anonymous.4open.science/r/SVRG_for_TD_learning-C688/Exp_result_grid_search_PD_SVRG.png). These results demonstrate, that  all PD SVRG algorithms exhibit geometric convergence, and 3 best performing  algorithms are those with highest $\sigma_\theta$, while value of $\sigma_w$ doesn't affect the performance that much. In addition, all 3 algorithms converge slower compare to TD-SVRG, probably because smaller learning rate and small batch size (it reevaluates mean path too often without necessity).
>
> For **GTD2** we also tried a set of values suggested in [Sutton2019], which are $\alpha \in (1/2, 1/4, 1/8, 1/16)$ and $\beta/\alpha \in (2,1,1/2,1/4)$, experiment setup file might be found [here](https://anonymous.4open.science/r/SVRG_for_TD_learning-C688/exp_setup_gtd2_grid_search.json). Results of 4 best performing methods compared against TD-SVRG are shown on this [figure](https://anonymous.4open.science/r/SVRG_for_TD_learning-C688/Exp_result_grid_search_GTD2.png). Similarly to previous experiment, these results show that value of $\beta$ doesn't affect the performance that much, and that the best performing value of $\alpha$ is 1/8, which, as we show in our paper, is optimal for this problem. All GTD methods exhibit sub-geometric convergence, while TD-SVRG converges geometrically.
>
> For **VRTD** we also tried a set of values close to [Xu2020], which are $\alpha = 0.1$ and batch sizes $M \in (500, 1000, 2000, 5000)$ (we added 5000 and removed smaller batch sizes), experiment setup file might be found [here](https://anonymous.4open.science/r/SVRG_for_TD_learning-C688/exp_setup_VRTD_grid_search.json). Results of these methods compared against TD-SVRG are shown on this [figure](https://anonymous.4open.science/r/SVRG_for_TD_learning-C688/Exp_result_grid_search_VRTD.png). Results demonstrate that, as expected, that batch size of 5000 is the best performing, but all of them are converge to some level of accuracy and oscillate near it, while TD-SVRG converges, with every step require larger batch sizes. This experiment shows disadvantage of practical VRTD: learning rate required to achieve theoretically guaranteed result is too small to be applied on practice, while practically applied values is hard to chose, i.e. if someone wants to run the experiment for given number of iteration, they don't know how to chose batch size such that it is not too small causing VRTD to converge to its potential best accuracy too fast (and later iteration will be wasted), but also it is not too big, so that VRTD will not converge to its best accuracy in given number of iterations.

---

> ### Author Response · Authors · 2022-11-30
> **Answer to updated review by Reviewer 158U**
>
> Here we address the reviewer's point about notation differences between this paper and the others we are comparing to. We will add a version of the following document to the appendix:
>
> https://anonymous.4open.science/r/SVRG_for_TD_learning-C688/Table1_comments_section.pdf
>
> This document describes the meaning of every symbol used in Table  1 containing the comparison of our results to the previous work. The version we will add to the paper will be slightly different, as we'll actually change the notation in Table 1 for greater clarity (we cannot edit the paper at this stage in the review process). We believe this answers the reviewer's question, but brings up another question we'll address next: given all these definitions, how do the various bounds in Table 1 compare?
>
> Relative to some of the previous work, our results are a straightforward improvement from $1/\lambda_{A}^2$ to $1/\lambda_A$ (and please don't miss the additional results we added in Section G of our manuscript showing a similar improvement in the Markov sampling case -- we did not add these to Table 1 before the deadline for paper editing passed). Moreover, we make the case that the scaling with $\lambda_A^{-1}$ going to be the most important factor: please see our replies to some of the other reviewers which contain some tables with numbers, showing that the various quadratic factors in Table 1 get astronomical very quickly.
>
> In other cases, the comparison is less straightforward. In particular, the first three lines of Table 1, summarizing previous work, have fairly complex expressions for the total sample complexities. Understanding all the quantities in these bounds in terms of the original policy evaluation problem is not super clear to us. Further, the various condition numbers are squared here, in contrast to the original SVRG in the context of convex optimization which scales linearly with condition number.
>
> This is what motivated us to write the present paper: we wanted to develop a TD-SVRG method which would replicate the guarantees of SVRG for the policy evaluation problem. Further, all of our condition numbers are of the standard matrix
>  $$ A = E [ (\phi(s) - \gamma \phi(s'))^T \phi(s)],$$
>
>  which appears throughout all analysis of TD; in the tabular case, this is just $A_{\rm tabular} = {\rm Diag}(\pi) (I - \gamma P)$ where $P$ is the probability transition matrix of the Markov chain and $\pi$ is its stationary distribution.
>
> We hope these remarks, along the notation section we have linked to above, both help readability as well as make the place of this paper in the context of the previous literature clearer.

---

### Author Response · Authors · 2022-11-18
**Answer to all reviews**

We thank the reviewers for their constructive comments! We would like to reiterate some of the key novel results of the paper and clarify our contribution:

We analyze the convergence of the SVRG technique applied to TD (TD-SVRG) in two settings: $(i)$ a pre-sampled trajectory of the  *Markov Decision Process (MDP)* (finite sampling), and $(ii)$ when states are sampled directly from the MDP (online sampling). Our contribution is threefold:

 * For the finite sample case we achieve significantly better results with simpler analysis. We are first to show that TD-SVRG has the same convergence rate as SVRG in the convex optimization setting with a pre-determined learning rate of 1/8.

 * For i.i.d. online sampling, we similarly achieve better results with simpler analysis. Similarly, we are first to show that TD-SVRG has the same convergence rate as SVRG in the convex optimization setting with a predetermined learning rate of $1/8$. In addition, for Markovian online sampling, we provide convergence guarantees that in most cases are better than state-of-the art results.

 * We are the first to develop theoretical guarantees for an algorithm that can be directly applied to practice. In previous works, batch sizes required to guarantee convergence were very large that made them impractical (see Subsection H.1) and grid search was needed to optimize learning rate and batch size values. We include experiments that show our theoretically obtained batch size and learning rate can be applied in practice and achieve geometric convergence.

We use blue colored text to highlight this and all other changes made by authors (except trivial typo and error fixes) in the rebuttal version of the paper. Blue section (subsection) title implies that whole section (subsection) is new. To fulfill page limit requirement, unbalanced case analysis and experiments with batched version of TD-SVRG were moved from the main body of the paper to the appendix.

---

### Decision · Program_Chairs · 2023-01-20

**Decision:**

Reject

**Justification For Why Not Higher Score:**

N/A

**Justification For Why Not Lower Score:**

N/A

**Metareview: Summary, Strengths And Weaknesses:**

The paper considers the Q-function evaluation via minimizing the mean-squared Bellman error, where the Q function is parameterized linearly. An SVRG technique is proposed to enhance the TD method so that the sample complexity has some improvement in the condition number.

The paper mainly has the following issues:

(1). When solving the finite-sum optimization, it is not correct to ignore the size of the full batch $N$. The authors seem to completely ignore the samples taken each time when SVRG restart with a full batch. Also, when analyzing finite-sum ERM problem, the authors seem to totally forget that this is an MDP problem instead of an abstract finite-sum problem. The authors do not show how the gap of the ERM problem relates to the mean-squared-error (MSE) or mean-squared-projected-Bellman-error (MSPBE) in the original Q-evaluation problem.

(2). Many notations are imprecise and possibly have some subtle mistakes hidden behind. For example, the notation $E_{ss'}$ is very vague and may affect the correctness of the analysis. For example, on page 3, when defining $\bar{g}(\theta)$, the $E_{ss'}$ is the over $\mu_\pi\times P$. However, see (4) for example, in the second inequality, $E_{ss'}$ is taken over the data distribution in the dataset $D$. Throughout the discussion, the authors use $f(\theta) = (\theta-\theta^*)^\top A (\theta-\theta^*)$ where $A$ is defined with the $E_{ss'}$ over the $\mu_\pi\times P$ distribution. But throughout the analysis like (4), the authors view $A$ as defined by the dataset $D$ distribution.  There are a lot of other impreciseness like this in the paper.

(3). Another concern is about the Assumption 3, dataset balancing. In the $\mu_\pi\times P$ distribution, $s$ and $s'$ indeed have the same distribution. However, when we move to a fixed dataset collected from the above distribution, even if $s_1 = s_N$, it is not so reasonable to think $s$ and $s'$ will have the same distribution. There should an unavoidable $O(1/\sqrt{N})$ statistical error. That is also why the confusion of $E_{ss'}$ under the true or empirical distribution can affect the correctness of the proof.

Nevertheless, the paper still has the merit of providing a simpler framework for analyzing this problem. We encourage the authors to resubmit the paper after clearing up the above-mentioned issues.

**Summary Of Ac-Reviewer Meeting:**

N/A